# Yielding Multi-Fold Training Strategy for Image Classification of Imbalanced Weeds

**Vo Hoang Trong [1] [iD], Yu Gwang Hyun [1], Kim Jin Young [1],\* and Pham The Bao [2],\* [iD]**

1   Department of ICT Convergence System Engineering, Chonnam National University, Gwangju 61186, Korea; hoangtrong2305@gmail.com (V.H.T.); sayney1004@gmail.com (Y.G.H.)
2   Faculty of Information Technology, Saigon University, Ho Chi Minh City 72710, Vietnam
\*   Correspondence: beyondi@jnu.ac.kr (K.J.Y.); ptbao@sgu.edu.vn (P.T.B.)

**Abstract:** An imbalanced dataset is a significant challenge when training a deep neural network (DNN) model for deep learning problems, such as weeds classification. An imbalanced dataset may result in a model that behaves robustly on major classes and is overly sensitive to minor classes. This article proposes a yielding multi-fold training (YMufT) strategy to train a DNN model on an imbalanced dataset. This strategy reduces the bias in training through a min-class-max-bound procedure (MCMB), which divides samples in the training set into multiple folds. The model is consecutively trained on each one of these folds. In practice, we experiment with our proposed strategy on two small (PlantSeedlings, small PlantVillage) and two large (Chonnam National University (CNU), large PlantVillage) weeds datasets. With the same training configurations and approximate training steps used in conventional training methods, YMufT helps the DNN model to converge faster, thus requiring less training time. Despite a slight decrease in accuracy on the large dataset, YMufT increases the F1 score in the NASNet model to 0.9708 on the CNU dataset and 0.9928 when using the Mobilenet model training on the large PlantVillage dataset. YMufT shows outstanding performance in both accuracy and F1 score on small datasets, with values of (0.9981, 0.9970) using the Mobilenet model for training on small PlantVillage dataset and (0.9718, 0.9689) using Resnet to train on the PlantSeedlings dataset. Grad-CAM visualization shows that conventional training methods mainly concentrate on high-level features and may capture insignificant features. In contrast, YMufT guides the model to capture essential features on the leaf surface and properly localize the weeds targets.

**Keywords:** imbalanced dataset; deep neural network; weeds classification; Grad-CAM

## 1. Introduction

With the application of deep neural network (DNN) models, many computer vision problems have achieved tremendous performance in tasks such as object classification [1], object segmentation [2], object detection [3], and object localization [4]. However, the success of DNN models may depend on the quality and distribution of the labels in the dataset. As shown in [5–9], a DNN model (and machine learning techniques) trained on an imbalanced dataset may show poor performance on minority labels. Cao et al. [10] mentioned that large-scale datasets might have long-tailed label distributions, meaning that dataset imbalances could become problematic when samples are collected in a large-scale domain. Common approaches to addressing this problem, such as those reviewed by Chawla in [11], include randomly oversampling minority or undersampling majority labels in data space or feature space. However, as the authors of [12,13] point out, these techniques may neglect potentially valuable data, sample unnecessary data, and risk losing relevant information. The authors of [13–15] solve the imbalance problem by assigning weights to classes. Others chose to combine multiple sampling strategies, such as [13,14,16]. In recent years, many methods have been proposed to deal with imbalanced datasets, such

as those using machine learning techniques [17,18], DNNs [19,20], generative adversarial networks (GANs) [21], or reinforcement learning (RL) [22].

Many approaches using conventional training of DNN models to solve the weeds classification problem using imbalanced weeds datasets, such as those of Trong et al. [23], who aggregated multiple DNN models and used the CNU Weeds dataset. The authors of [24–32] developed and proposed novel models and techniques and applied them to the PlantSeedlings and PlantVillage datasets. In reality, weeds image datasets may have non-uniform distributions due to the relative prevalence of various species, some of which are common and diverse, while others are rare and identical. Hence, training a DNN model by selecting a batch of random samples from an entire training set makes the model robust to majority species, but sensitive to minority species, as Johnson and Khoshgoftaar pointed out in their survey [33].

To reduce this bias, we proposed the yielding multi-fold training (YMufT) strategy, which trains the DNN model by arbitrarily dividing the training data into multiple folds and then trains the model on each of these folds consecutively. Furthermore, samples of minority species are trained more often than those of majority species so that the model pays more attention to minority species. The MCMB procedure assures this specification to determine the number of samples of each species selected and passed on each consecutive fold. In addition, we propose a formula to determine the number of training loops and training periods in YMufT. This formula aims to ensure that the number of loading training images in the YMufT strategy is approximately the same as, or less than, that in conventional training methods.

We explore our strategy on four weeds datasets, containing two large datasets (CNU and large PlantVillage) and two small datasets (PlantSeedlings and small PlantVillage). Both types of datasets have plain and in-the-wild weeds datasets. Experiments are done using three DNN models (Mobilenet, Resnet, and NASNet mobile) and the results are compared to those of the conventional training methods. We find that YMufT runs faster during the training period and is quicker to converge when training on the validation set, even though both approaches have the same training configurations and approximate training steps. The evaluation shows that training the DNN model using the YMufT strategy results in comparable performance to conventional training strategies on large datasets and higher performance on small datasets. Specifically, YMufT increases the F1 score in the NASNet model to 0.9708 on the CNU dataset and 0.9928 by using Mobilenet model training on large PlantVillage dataset. On the small PlantVillage dataset, Mobilenet achieves the highest performance of 0.9981 in accuracy and 0.9970 in F1 score. On the PlantSeedlings dataset, Resnet shows the highest accuracy (0.9718) and F1 score (0.9689). Grad-CAM visualization shows that YMufT guides the model to capture essential features of the leaf surface and correctly localize weeds targets. Conventional training methods mainly concentrate on high-level features and may capture insignificant features.

In summary, our main contributions are:

We propose a YMufT strategy to train the DNN model on imbalanced datasets. Using this strategy, the model is more generalized, performs better, and is less time-consuming during the training steps than conventional training methods.

We propose the MCMB procedure to select samples of each species to form a fold used to train the model. This fold has a uniform distribution, or at least much less imbalance than the distribution of species across the training set. This procedure also shows that the model pays more attention to minority species, reducing bias towards majority species.

Experiments show that YMufT improves performance on minority species while maintaining performance on majority species. In particular, YMufT outperforms conventional training methods on small datasets in terms of overall performance.

Grad-CAM visualization reveals that a model trained by the YMufT strategy tends to focus on microscopic features on the leaf surfaces, while the conventional trained model tends to concentrate on high-level features of the leaf.

## 2. Related Works

Kang et al. [17] and Lin et al. [22] noted that dealing with the imbalance problem by undersampling may increase computational complexity and decrease performance. Several approaches aimed to combine multiple sampling techniques, such as that of Gonzalez et al. [13], who proposed a set of new sampling techniques and applied inside monotonic chains to preserve the monotonicity of the datasets and to improve performance on minority classes. Liu et al. [14] simultaneously combined ensemble learning, evolutionary undersampling, and multiobjective feature selection. This combination resulted in an imbalanced classification method called Genetic Under-sampling and Multiobjectie Ant Colony Optimization based Feature selection, which efficiently solved the imbalanced classification problem. Nejatian et al. [16] proposed modified-bagging by combining multiple poor classifiers similar to the decision tree method. This algorithm is suitable for an imbalanced dataset in which the samples of the minority classes are much less frequent than those of the majority class.

Another approach aimed at weight classes depends on the classes' contribution to the total population. Zhang et al. [12] assigned different misclassification costs for different classes based on the training data. Without prior domain knowledge, these costs were optimized using the adaptive differential evolution algorithm, which was then applied to the deep belief network. Khan et al. [34] proposed a cost-sensitive DNN to learn robust feature representation for all classes. They incorporated class-dependent costs during model training. They automatically set these costs using statistics. Shu et al. [15] proposed a one-hidden layer multi-layer perceptron (MLP) to learn weights directly from the data. During the training, they used a small, unbiased validation set to choose the optimal training parameters.

Besides applying sampling techniques, the use of the area under the receiver operating characteristic (ROC) is also a prominent approach to classification in the presence of imbalanced class distribution. The effectiveness of a ROC strategy depends on the classification threshold. Zou et al. [35] proposed a sampling-based threshold auto-tuning method to identify the optimal classification threshold. Their approach improved classification performance over other commonly employed methods.

Using support vector machine (SVM) is a common method for classification problems in conventional machine learning, but an SVM loses its effectiveness on large-scale imbalanced datasets. Kang et al. [17] proposed a weighted undersampling scheme to improve SVM performance. This scheme assigns weights to majority classes based on their Euclidean distance to the hyperplane. By grouping samples based on their weights, this scheme reduces the number of majority classes. Lemnaru and Potolea [18] studied possible solutions to the class imbalance problem. They analyzed many standard classification algorithms, such as decision trees, Bayesian methods, and SVM. They found that none of these algorithms help all datasets, but MLP was the most robust to the imbalance problem, and SVM performed well on artificial data.

With the development of deep learning, many methods applied this technique to solve imbalance problems. Jia et al. [19] applied a deep normalized convolutional neural network (CNN) to imbalanced fault classification of machinery. Using a neuron activation maximization algorithm to analyze kernels in the convolutional layers, they found that these kernels behave like filters—the deeper the layer, the more complex these kernels. Dong et al. [20] formulated a class imbalanced deep learning model to train a model on an imbalanced dataset. They designed the model to minimize the dominance of majority classes using batch-wise mining of complex samples. They also proposed a class rectification loss regularization algorithm for minority class incremental rectification. Mullick et al. [21] argued that oversampling techniques such as synthetic minority over sampling technique or deep oversampling framework could not be applied to an end-to-end deep learning system. Thus, they re-approached oversampling techniques by proposing an end-to-end feature-extraction-classification framework consisting of a convex generator, a

multi-class classifier network, and a real/fake discriminator to generate new samples from minority classes.

An RL approach was proposed by Lin et al. [22]. They argued that conventional classification algorithms fail when the data distribution is imbalanced. Using deep Q-learning, they formulated this problem as a sequential decision-making process. The agent performs a classification action. The environment evaluates this action and returns a reward to the agent such that the more minor the class, the larger the reward. This results in a model that pays more attention to the minority classes.

## 3. Datasets

We analyze the YMufT strategy and compare it with conventional training methods on small and large weeds datasets. In addition, we select in-the-wild and plain datasets to further analyze the efficiency of the YMufT strategy over a variety of weeds datasets. The Chonnam National University (CNU) weeds dataset helps us evaluate YMufT in comparison to conventional methods on a large and in-the-wild dataset. In contrast, the small PlantVillage dataset is used to examine the robustness of the two training approaches on a small and plain dataset. Another small dataset, the PlantSeedlings dataset, is used to investigate behavior of the DNN models on an in-the-wild dataset. Finally, the large PlantVillage dataset is used to helps us consider the benefit of YMufT on a large and plain dataset.

### 3.1. PlantSeedlings Dataset

The PlantSeedlings dataset [36] contains images of plant seedlings placed on Styrofoam boxes. This dataset contains 5539 images of 12 species. Examining the distribution of species in this dataset (shown in Figure 1) shows that the dataset is imbalanced. Common Chickweed is the species with the largest number of images (713 images), while common wheat has the fewest images (253 images). Figure 2 shows examples of samples of all 12 species.

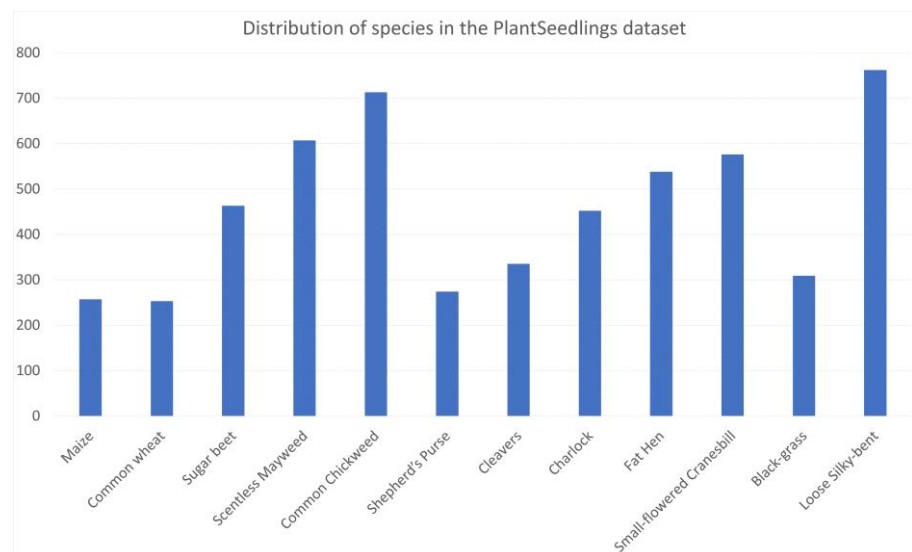

**Figure 1.** Distribution of species in the PlantSeedlings dataset.

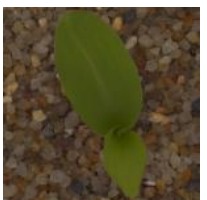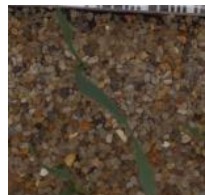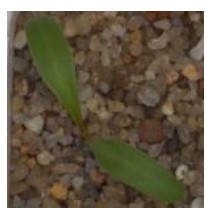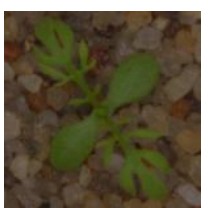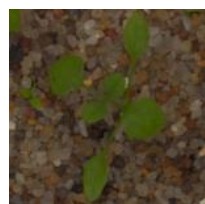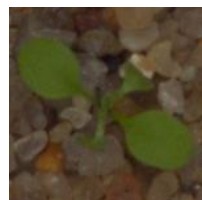

**Figure 2.** *Cont*.

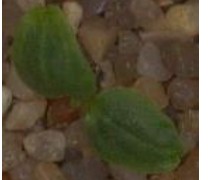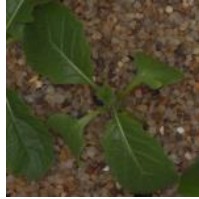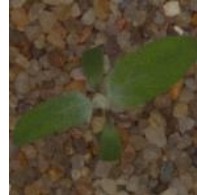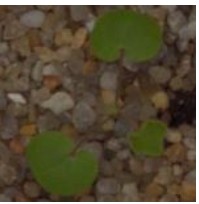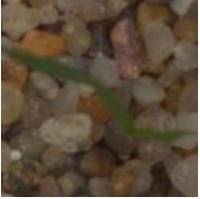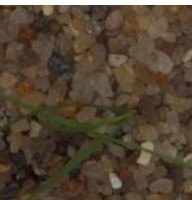

**Figure 2.** Example images of the 12 species in the PlantSeedlings dataset. The order of species (left to right, top to bottom) corresponds to the labels (from left to right) in Figure 1.

### 3.2. Small PlantVillage Dataset

The PlantVillage dataset [37] consists of 39 classes separated by species and disease. This dataset has two versions: with and without augmentation. The non-augmentation version is referred to as the small dataset. This dataset has 55,447 images in total. Figure 3 shows the distribution of classes, which is clearly imbalanced. *Orange___Haunglongbing_ (Citrus_greening)* has the highest number of images (5507 images), while *Potato___healthy* has the fewest images (152 images). Figure 4 shows examples of each of the 39 classes.

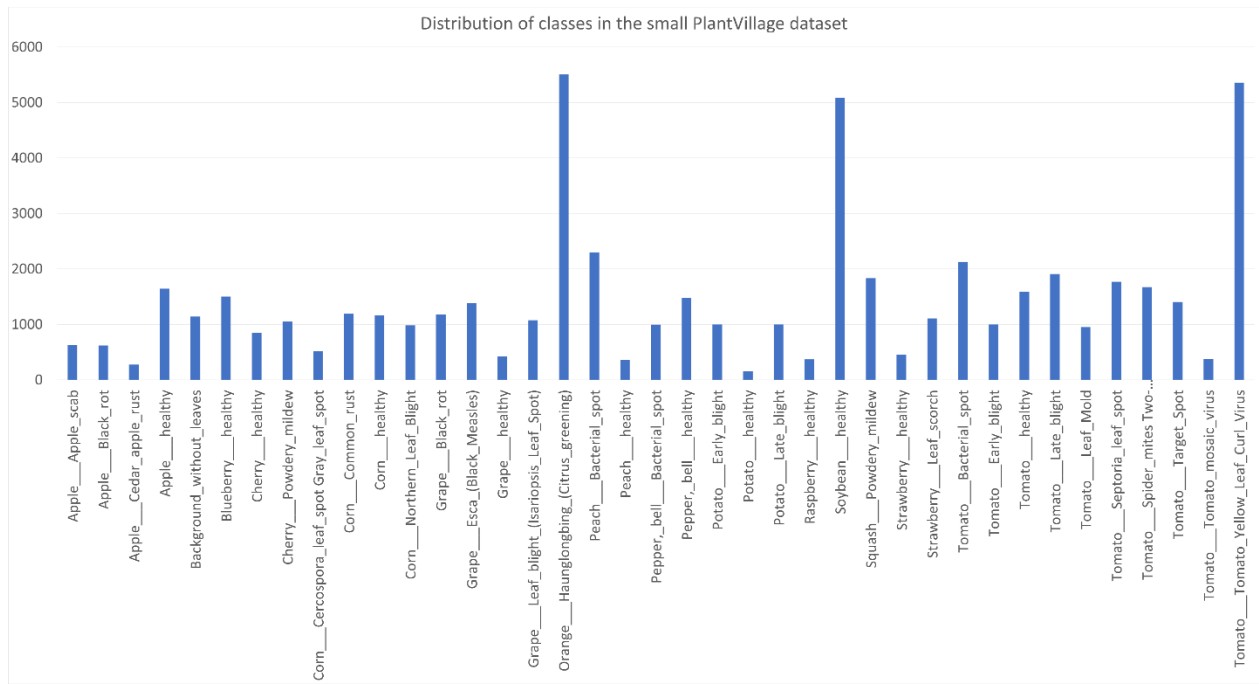

**Figure 3.** Distribution of classes in the small PlantVillage dataset.

### 3.3. CNU Weeds Dataset

This dataset contains 208,477 images of 21 species produced by CNU, Gwangju, Republic of Korea. Images of weeds were captured and collected by Korean plant taxonomists working on farms and fields in the Republic of Korea with high-definition resolution cameras. Figure 5 shows example images from the CNU Weeds dataset, and Figure 6 shows the distribution of species in this dataset. This dataset is imbalanced. The species with the largest number of images is *Galinsoga quadriradiata* Ruiz & Pav. (24,396 images), while the species with the smallest number of images is *Bidens bipinnata* L. (804 images).

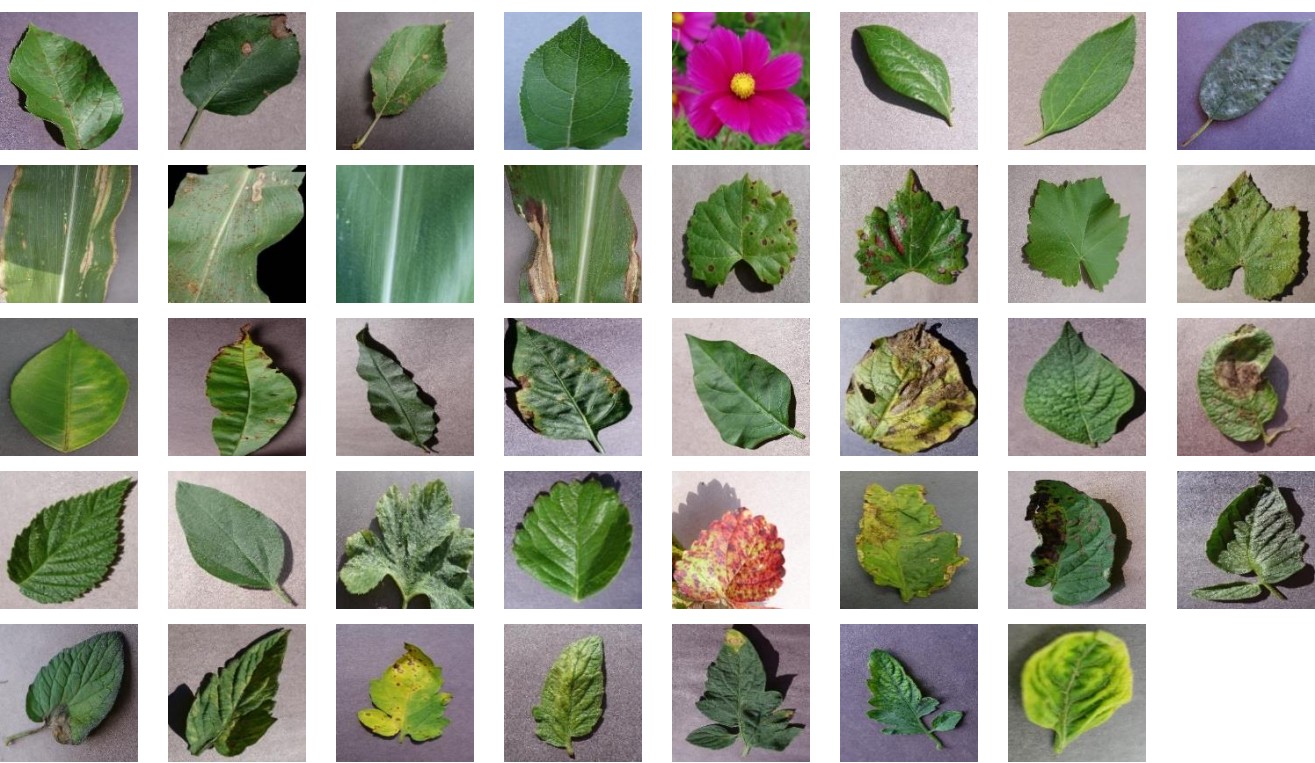

**Figure 4.** Example images of each of the 39 classes in the small PlantVillage dataset. The order of classes (left to right, top to bottom) corresponds to the labels (from left to right) in Figure 3.

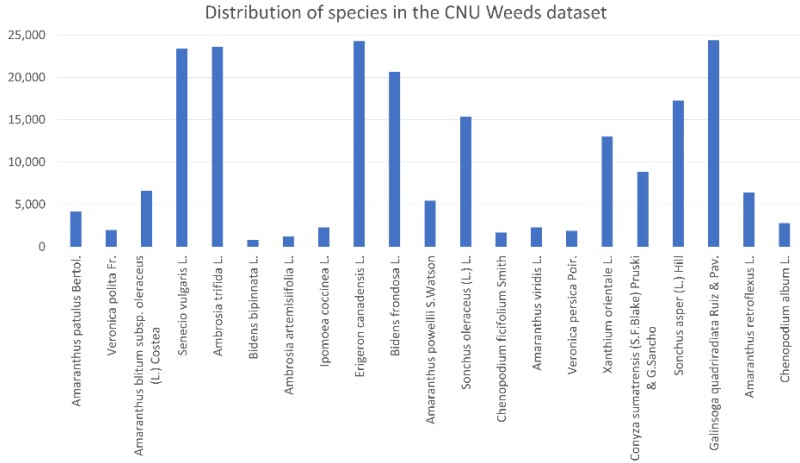

**Figure 5.** Distribution of species in the CNU weeds dataset.

### 3.4. Large PlantVillage Dataset

The large PlantVillage dataset is built off of the small PlantVillage dataset, with the following six data augmentation techniques applied to increase the size of the dataset: flipping, gamma correction, noise injection, PCA color augmentation, rotation, and scaling. The large PlantVillage dataset has 61,485 images. Figure 7 shows the distribution of the 39 classes in the large PlantVillage dataset. Like the small PlantVillage dataset, this dataset is imbalanced, and *Orange___Haunglongbing_(Citrus_greening)* is the class with the largest number of images (5507 images). Seventeen classes have the smallest number of images (1000 images each).

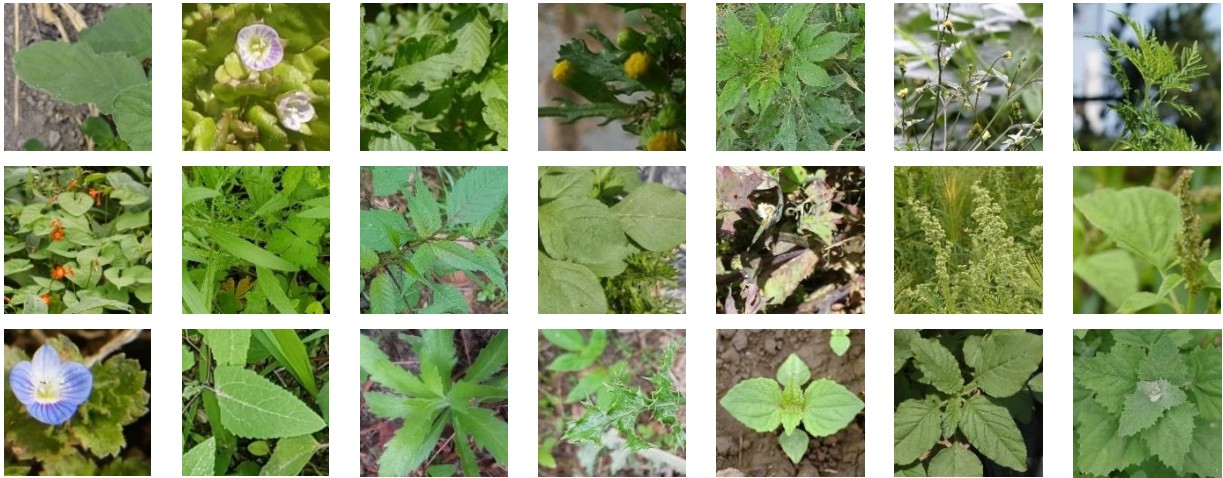

**Figure 6.** Examples of the 21 species in the Chonnam National University (CNU) weeds dataset. The order of species (left to right, top to bottom) corresponds to the labels (from left to right) in Figure 5.

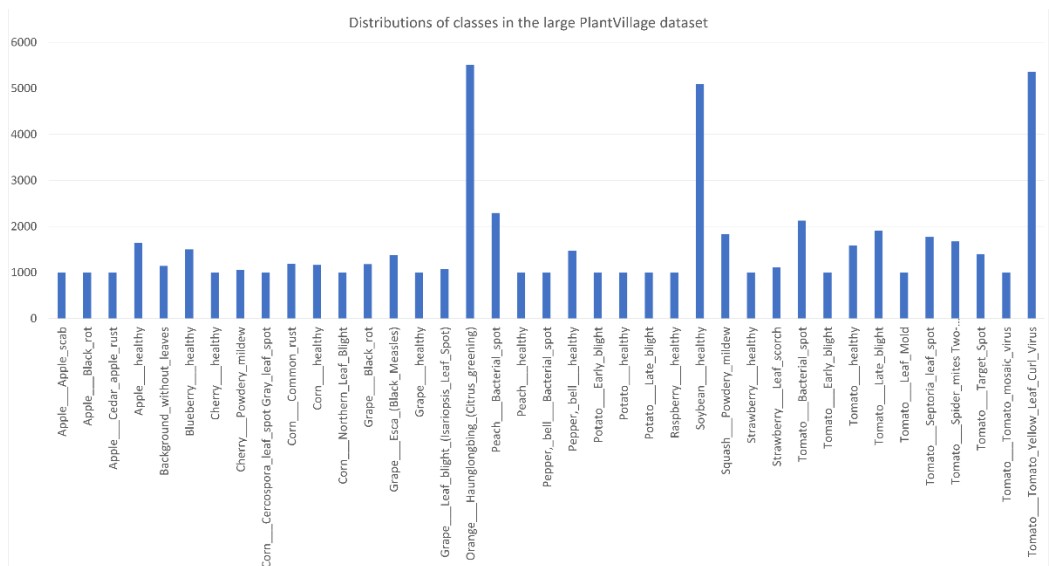

**Figure 7.** Distribution of classes in the large PlantVillage dataset.

## 4. Methodology

Given a training set and a DNN model, the conventional training methods first train the whole training model with several epochs. In each epoch, the samples are randomly divided into an equal number of batches and each batch is fed to the model. No sample appears in two different batches. After the model is trained on every epoch, it is validated on the validation set. With this strategy, the model tends to encounter majority species more often than minority species, making the model robust to majority species since it can capture the whole variety of features in the majority samples. Otherwise, the lack of samples of minority species would make the model less focused and cause difficulties learning about the general characteristics of these species.

Our proposed strategy (YMufT) solves this problem by dividing the samples in the training set into multiple folds. Minority species are presented to the model more often than majority species. The smaller the number of samples, the more times the related species will be learned. This division strategy reduces the bias of the DNN model towards the majority species. We define the balance error (BE) to measure the imbalance of the dataset (Definition 1).

**Definition 1 (Balance Error).** *A dataset $\mathcal{D}$ has c species, and each species $y_i$ has $n_i$ samples. $n_{max}$ and $n_{min}$ are the largest and smallest number of samples in a species, respectively. We calculate the Balance Error BE of $\mathcal{D}$ using Formula (1) below. The lower the value of $BE_{\mathcal{D}}$, the greater the balance of $\mathcal{D}$.*

$$BE_{\mathcal{D}} = \frac{n_{\max} - n_{\min}}{\sum_i n_i} \tag{1}$$

Consider the number of samples in a species as a discrete random variable. By this definition, $BE_{CNU}$ of the CNU Weeds dataset is 0.11316. Only five species have more than the median number of samples (18,974), and the standard deviation around this median is 12,557.854. In the large PlantVillage dataset, $BE_{lPV} = 0.0733$, and 11 classes have more samples than the median (1546.5). The standard deviation is 1135.824. In the small PlantVillage dataset, while $BE_{sPV} = 0.0966 > BE_{lPV}$, only 10 classes have more than the median number of samples (1618). The standard deviation around this median is 1254.96. In the PlantSeedlings dataset, $BE_{PS} = 0.0920$, and five species have more than the median number of samples (500.5). The standard deviation is 97.62. Considering the uniform distribution as the standard distribution for a balanced dataset, we can measure the relative entropy from the distribution of a dataset to the uniform distribution. Equation (2) is the formula used to calculate the relative entropy from $P$ to $Q$, where $P$ and $Q$ are two discrete probability distributions and $\chi$ is a set of species in a dataset.

$$D_{KL}(P \parallel Q) = \sum_{x \in \chi} P(x) \log_2 \left( \frac{P(x)}{Q(x)} \right) \tag{2}$$

Assume that $P$ is the probability distribution of a dataset with $c$ species, and $Q$ is a uniform distribution. We can rewrite (2) to (3).

$$D(P) = \log_2(c) + \sum_{x \in \chi} P(x) \log_2(P(x)) \tag{3}$$

Using (3) to calculate the relative entropy of the distribution of a dataset to the uniform distribution, we obtain $D(P_{CNU}) = 0.5649$, $D(P_{lPV}) = 0.2597$, $D(P_{sPV}) = 0.415$, and $D(P_{PS}) = 0.0998$ corresponding to the CNU Weeds, large PlantVillage, small PlantVillage, and PlantSeedlings datasets, respectively. This shows that the distribution of the PlantSeedlings dataset is close to uniform, while the CNU Weeds dataset is quite imbalanced.

We define a set $\mathcal{Y}$ consisting of the number of samples of each species. This set has $c$ elements, in which the $i^{th}$ element is $n_i$. The notation of the $n^{th}$ maximum and minimum value in $\mathcal{Y}$ is $\max(\mathcal{Y}, n), \min(\mathcal{Y}, n)$.

Instead of training the DNN model $M$ directly on $\mathcal{D}$, we create a fold $f$ and train the model on this fold. Folds are established by randomly collecting $\min(\mathcal{Y})$ samples of each species and placing them in $f$. However, if the difference $\epsilon$ between the minimum and nth minimum value in $\mathcal{Y}$ is small, then we collect $\min(\mathcal{Y}, n)$ samples of each species and place them into $f$, except we take all samples from a species containing a number of samples between $\min(\mathcal{Y}, 1)$ and $\min(\mathcal{Y}, n - 1)$.

These perceptions lead us to come up with the MCMB procedure to divide samples into folds. First, we determine Min-Class in $\mathcal{Y}$, which is $MC$argmin$(\mathcal{Y}, 1)$. Then, we form a set $S = \{n_i \in \mathcal{Y} : n_i \leq \epsilon + \mathcal{Y}[MC]\}$ containing elements in $\mathcal{Y}$ that are close to MC and determine Max-Bound of $S$, which is $MB$max$(S)$. For all species, we take $\min(MB, n_i)$ samples of species $i$ in $\mathcal{Y}$ and place them in $f$. In practice, $\epsilon$ is determined by an excess ratio $k \in [0, 1]$ to $\mathcal{Y}[MC]$, which is $k \cdot \mathcal{Y}[MC]$. This ratio aims to guarantee that the number of samples in the "Max-Bound" species cannot be more than 2 times greater than $MC$ to avoid high imbalance distribution on $f$.

Algorithm 1 shows the algorithm of the YMufT strategy. The folds division process proceeds until all samples are being divided. In this process, $A_{\text{temp}}$ is a list containing non-divided samples, and $\mathcal{Y}$ consists of the number of samples in $A_{\text{temp}}$. The MCMB

procedure is only applied on $A_{\text{temp}}$. After establishing the 1st fold, dividable samples are removed from $A_{\text{temp}}$. Additionally, all samples in species $MC_1$ are being divided, so $\mathcal{Y}[MC_2] = 0$. In the subsequent division, we select $\min(\mathcal{Y}[i], MB_2)$ samples for each species $i$, except species $MC_2$, for which we select $\min(B[i], MB_2)$ samples from $A$, where $A$ is a list containing all samples in $\mathcal{D}$ and $B$ is a set containing the number of samples in species $y_i$ in the original training dataset.

Generally, given the $d^{th}$ fold division process, assume the MCMB procedure returns $M$. We randomly select $\min(\mathcal{Y}_d[i], MB_d)$ if species $i$ has non-divided samples or $\min(B[i], MB_d)$ if all samples in species $i$ have already been divided. Figure 8 illustrates the YMufT strategy with five species. With $k = 0.5$, YMufT divides this data into four folds.

By applying the MCMB procedure, all samples of minority species are divided into the first few folds. We randomly re-select these samples at the next folds while choosing non-divided samples of the majority species and appending them to those folds. This action means that samples of minority species are selected more frequently, and the model may focus on learning the feature characteristics of those species, hence reducing the degree of bias toward majority species.

---

**Algorithm 1.** Algorithm of the yielding multi-fold training (YMufT) strategy.

---

$\text{YMufT}(\mathcal{D}, c, k)$

input　　$\mathcal{D}$ : A dataset has samples $x_i$ with corresponding species $y_i$. $c$: Number of species in $\mathcal{D}$. $k$: An excess ratio, $k \in [0, 1]$.

output　 List of folds $\mathcal{F}$

Initialize $A$ has $c$ rows, $A[i] = \left\{ \left( x_j^i, y_j^i \right) \big| x_i \in y_i \right\}$

**Step 1**　Initialize $B$, $B[i] \leftarrow |y_i|$.

Initialize $A_{\text{temp}} \leftarrow A$, $\mathcal{Y} \leftarrow B$, an empty list $\mathcal{F}$.

**While** $\exists m \in \mathbb{N}, \mathcal{Y}[m] > 0$ **do**:

　Initialize an empty fold $f_i$

　$MC \leftarrow \overset{+}{\text{argmin}}(\mathcal{Y})$　　　　　　　　//Select the species has the smallest positive value in $\mathcal{Y}$.

　$\epsilon \leftarrow k \cdot \mathcal{Y}[MC]$　　　　　　　　　　　//Determine the maximum possible boundary.

　$inbou \leftarrow \{\mathcal{Y}[\cdot] : \mathcal{Y}[\cdot] \leq \epsilon + \mathcal{Y}[MC]\}$　//List of species that lay in the boundary.

　$MB \leftarrow \max(inbou)$　　　　　　　　　　//Select the maximum value.

　**For** $i1$ **to** $c$ **do**:

　　**If** $\mathcal{Y}[i] > 0$ :　　　　　　　　　　$nt = \min(\mathcal{Y}[i], MB)$

**Step 2**　　　　　　　　　　　　　　　　　　　　//Randomly select $nt$ samples from $A_{\text{temp}}[i]$.

　　　$S_i \subseteq A_{\text{temp}}[i], |S_i| = nt$

　　　$f_i.\text{append}(S_i)$

　　　$A_{\text{temp}}[i] \leftarrow A_{\text{temp}}[i] \backslash S_i$　　　//Delete these $nt$ samples in $A_{\text{temp}}[i]$.

　　　$\mathcal{Y}[i] \leftarrow \mathcal{Y}[i] - nt$

　　**Else**:　　　　　　　　　　　　　　　　　$nt = \min(B[i], MB)$

　　　　　　　　　　　　　　　　　　　　　//Randomly select $nt$ samples from $A[i]$.

　　　$S_i \subseteq A[i], |S_i| = nt$

　　　$f_i.\text{append}(S_i)$

　$\mathcal{F}.\text{append}(f_i)$

**Step 3**　Return $\mathcal{F}$.

---

**Figure 8.** Illustration of the YMufT strategy, with $k = 0.5$. The original dataset contains 5 species: A, B, C, D, and E.

Assume a list $\mathcal{F}$ contains $q$ folds. We have the following two definitions:

**Definition 2.** *A training period is when the model trains all q folds consecutively from the first fold $f_1$ to the last fold $f_q$.*

**Definition 3.** *A training loop is a process in which the model trains a fold on a finite loop.*

In the YMufT training strategy, we argue that training a model with $\alpha$ loops on $P$ periods is not beneficial because the number of samples in each fold is much smaller than $N_{train}$, leading to overfitting if $\alpha$ is set too high. After capturing the characteristics of the samples, the model may converge quickly in later periods, leading to poor generalizability. To deal with this problem, we reduce $\alpha$ in later periods. We assign a finite sequence $(\alpha_k)_{k=1}^{P}$, indicating the number of training loops $\alpha_k$ in the $k^{th}$ training period, and ensure that $\alpha_1 \geq \alpha_2 \geq \ldots \geq \alpha_P$. After the completion of $P$ training periods, the total number of times the loading samples are calculated is given by Formula (4)

$$N_{YMufT} = \sum_{k=1}^{P} \left( \alpha_k \cdot \sum_{i=1}^{q} \left( \left\lfloor \frac{|f_i|}{nb} \right\rfloor \cdot nb \right) \right) \tag{4}$$

$$\approx \sum_{k=1}^{P} \left( \alpha_k \cdot \sum_{i=1}^{q} (|f_i|) \right) \tag{5}$$

$$= \sum_{k=1}^{P} \alpha_k \cdot \sum_{i=1}^{q} (|f_i|) \tag{6}$$

where $nb$ is the number of batches and $|f_i|$ indicates the number of samples in fold $f_i$. In the conventional training method, if we train the model on $\mathcal{D}$ for *eps* epochs and the number of batches is $nb$, the total number of times the loading samples are calculated is given by Formula (7).

$$N_{conve} = \left\lfloor \frac{|\mathcal{D}|}{nb} \right\rfloor \cdot nb \cdot eps \approx |\mathcal{D}| \cdot eps \tag{7}$$

To ensure that the YMufT training strategy results in faster training than conventional training methods, we solve inequality (8).

$$N_{YMufT} \leq N_{conve} \tag{8}$$

$$\Leftrightarrow \sum_{k=1}^{P} \alpha_k \cdot \sum_{i=1}^{q} (|f_i|) \leq |\mathcal{D}| \cdot eps \tag{9}$$

Since we can determine $q, |f_i|, |\mathcal{D}|$ and the number of epochs *eps* used in conventional training, we only need to define the sequence $(\alpha_k)_{k=1}^{P}$ that satisfies inequality (8). If we choose this sequence as consecutive natural numbers beginning with 1, inequality (8) becomes

$$\sum_{k=1}^{P} \alpha_k \cdot \sum_{i=1}^{q} (|f_i|) \leq |\mathcal{D}| \cdot eps \tag{10}$$

$$\Leftrightarrow \frac{P(P+1)}{2} \cdot \sum_{i=1}^{q} (|f_i|) \leq |\mathcal{D}| \cdot eps \tag{11}$$

$$\Leftrightarrow P^2 + P - \frac{2|\mathcal{D}| \cdot eps}{\sum_{i=1}^{q} (|f_i|)} \leq 0 \tag{12}$$

Based on Vieta's formulas, the left-hand side in inequality (12) has one positive solution. We select the natural number $P \in \left[ 1, \frac{-1+\sqrt{\Delta}}{2} \right]$ to satisfy (8), in which

$$\Delta = 1 + \frac{8|\mathcal{D}| \cdot eps}{\sum_{i=1}^{q} (|f_i|)} \tag{13}$$

To assure the maximum possible value of $P$ is at least 1, we need to solve inequality (14):

$$\frac{-1 + \sqrt{\Delta}}{2} \geq 1 \tag{14}$$

$$\Rightarrow 1 + \frac{8|\mathcal{D}| \cdot eps}{\sum_{i=1}^{q}(|f_i|)} \geq 9 \tag{15}$$

$$\Leftrightarrow eps \geq \frac{\sum_{i=1}^{q}(|f_i|)}{|\mathcal{D}|} \tag{16}$$

which means that the number of epochs in a conventional training method must be greater than or equal to the ratio between the total number of samples in $q$ folds and the total number of samples in the training set.

## 5. Experiments

### 5.1. Performance Metrics

We applied the metrics described in [23] to evaluate the performance of the conventional and YMufT strategies for training of a DNN model on an imbalanced dataset. Suppose the evaluation dataset $D$ contains $m$ images of $c$ species. Assume $R_i \subset D$ is the set of images classified as species $c_i$, then

- True Positive ($TP_i$): The number of images in $R_i$ that are classified correctly.
- False Positive ($FP_i$): The number of images in $R_i$ that are classified incorrectly.
- False Negative ($FN_i$): The number of images of species $c_i$ that are incorrectly classified as not being $c_i$.

We define the overall performance using four metrics: Accuracy, Precision, Recall, and F1 score.

- Accuracy: The percentage of images in $D$ that are correctly classified.

$$\text{Accuracy} = \frac{\sum_{i=1}^{c} TP_i}{m} \tag{17}$$

- Precision: The average percentage of images predicted to belong to species $c_i$ that are correctly classified, across all $c$ species.

$$\text{Precision} = \frac{1}{c}\sum_{i=1}^{c}\frac{TP_i}{TP_i + FP_i} \tag{18}$$

- Recall: The average percentage of images in $c_i$ that are correctly classified across all $c$ species.

$$\text{Recall} = \frac{1}{c}\sum_{i=1}^{c}\frac{TP_i}{TP_i + FN_i} \tag{19}$$

- F1 score: The harmonic means of precision and recall. This metric is suitable for measuring the performance of training strategies on an imbalanced dataset.

$$\text{F1 score} = 2 \cdot \frac{\text{Precision} \cdot \text{Recall}}{\text{Precision} + \text{Recall}} \tag{20}$$

In addition, we measured precision and recall on every species to estimate the behavior of minority and majority species.

### 5.2. Training of DNN Models

We trained the DNN models using the Keras library on an Ubuntu 16.04.5 LTS Linux server, Intel(R) Core(TM) i9-7900X CPU @ 3.30 GHz, 125 GB RAM. The graphics processing unit is a 12 GB NVIDIA TITAN V with CUDA 10.1. We selected 3 DNN models for the experiment. These models have different architectures, and capture species characteristics in different ways.

- Mobilenet [38]. This model architecture uses depthwise and pointwise convolution to learn an object's features. There are fewer parameters in the model than in traditional convolution operators. Mobilenet is the lightest of the 3 DNN models. We used Mobilenet version 1, which has nearly 3.5 million trainable parameters.
- Resnet [39]. This is a deep residual learning model in which a shortcut connection is added between two blocks of convolutional layers, allowing information from one layer to flow directly to another layer. We used the 50-layer Resnet model that has over 24 million trainable parameters.
- NASNet [40]. This model architecture is generated by using Neural Architecture Search to build a network from the ImageNet dataset. We used the mobile version of NASNet, which has over 4.5 million trainable parameters.

We applied transfer learning, and used parameters trained on the ImageNet dataset as the initial parameters to train the model. We fine-tuned all three models by adding a fully connected layer of length 256, batch normalization, ReLU, and a Softmax layer. The input RGB image for the three models was $128 \times 128$, normalized to the range $[0, 1]$. Stochastic Gradient Descent was used as the optimization method with a learning rate of 0.001. We applied these model configurations on both strategies to enable a fair comparison between the conventional and YMufT strategies, except that the batch size varied depending on the model and dataset. Table 1 show the batch sizes for each model and dataset, which were applied for both training approaches.

**Table 1.** Batch size according to model and dataset.

|  | CNU | PlantVillage (Large) | PlantVillage (Small) | PlantSeedlings |
|---|---|---|---|---|
| Mobilenet | 128 | 32 | 32 | 128 |
| Resnet | 32 | 16 | 16 | 32 |
| NASNet | 16 | 8 | 8 | 16 |

We evaluated model performance on two small datasets (small PlantVillage and PlantSeedlings) by applying 5-fold cross-validation and 2 data augmentation techniques, random rotation, and random zoom. We used the YMufT strategy and divided the training set into folds. Due to the small number of samples in each fold, we duplicated the images in each fold four times in PlantSeedlings and three times in the small PlantVillage dataset to avoided overfitting when training the model using those folds. On two large datasets (CNU Weeds and large PlantVillage), we randomly selected 60% of each species' images for training, 20% for validation, and 20% for testing. We used no data augmentation techniques in the training set.

We applied a sequence of consecutive numbers $(\alpha_k)_{k=1}^{P}$ as the training loop on $P$ periods, which required us to select a value of $P$ that satisfied inequality (16). Table 2 shows the total number of samples in the folds (2nd column) and the training set (3rd column) with $k = 0.5$. The ratio in the 4th column indicates that use of $eps \geq 8$ when training the model in the conventional training method guarantees that $P \geq 1$. We trained the DNN models using the conventional training method on 50 epochs in CNU Weeds and PlantVillage, and 100 epochs on the PlantSeedlings dataset. We chose the maximum possible natural number $T$ that satisfies inequality (16), as shown in Table 3.

**Table 2.** Total number of samples in the folds and training set.

|  | $\sum_{i=1}^{q}(|f_i|)$ | $|\mathcal{D}|$ | $\frac{\sum_{i=1}^{q}(|f_i|)}{|\mathcal{D}|}$ |
|---|---|---|---|
| CNU | 271,713 | 125,081 | 2.17 |
| PlantVillage (large) | 80,124 | 36,895 | 2.17 |
| PlantVillage (small) | 329,721 | 44,360 | 7.43 |
| PlantSeedlings | 28,296 | 4431 | 6.39 |

**Table 3.** Number of training periods, *P*.

|   | CNU | PlantVillage (Large) | PlantVillage (Small) | PlantSeedlings |
|---|-----|----------------------|----------------------|----------------|
| *P* | 6 | 6 | 3 | 5 |

In the conventional training method, we validated the model on every epoch. In YMufT, we made every training loop through all periods. The number of times the model is validated in YMufT is calculated using expression (21). Table 4 shows the number of validations in the conventional and YMufT strategies. As shown in this table, the YMufT strategy required more time to validate the model than the conventional training strategy, except on the PlantSeedlings dataset.

$$q \cdot \frac{P(P+1)}{2} \tag{21}$$

**Table 4.** Number of times the model was validated in the conventional and YMufT strategies.

|   | CNU | PlantVillage (Large) | PlantVillage (Small) | PlantSeedlings |
|---|-----|----------------------|----------------------|----------------|
| Conventional | 50 | 50 | 50 | 100 |
| YMufT | 315 | 231 | 156 | 90 |

### 5.3. Computational Complexity

Two factors affect the processing time of both approaches: The training and validation time of the model. Assume that the model finishes training, without validation, at time $T_t$, and the time it takes to validate the model is $T_v$. We estimated the total time $T$ for training and validation of the model using Formula (22)

$$T = T_t + vT_v \tag{22}$$

where $v$ is the number of times the model was validated. In the conventional training method, $v$ equals the number of epochs, while in YMufT, $v$ was calculated using Formula (21).

Figure 9 shows the number of times, $T_v$, the model was validated. Only Resnet on the CNU Weeds dataset ran faster than Mobilenet. On the other datasets, Mobilenet was the fastest at validating a model, followed by Resnet and NASNet. Figure 10 compares $T_t$ and $T$ when training a model using the conventional method and YMufT strategy. In every case, YMufT required slightly less time to train a model than the conventional method. However, the increase in the number of validations meant that the YMufT strategy took more time to complete the training process, except on the PlantSeedlings dataset.

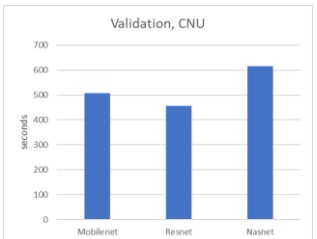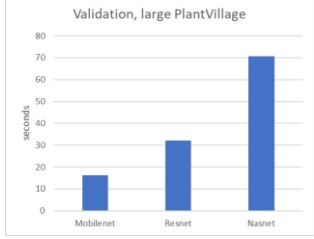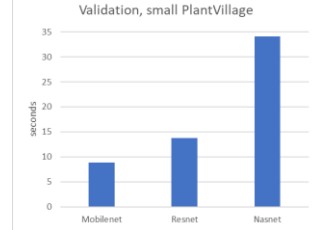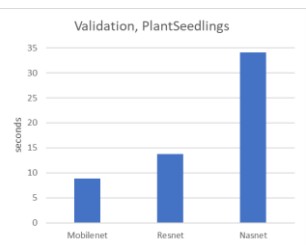

**Figure 9.** Total validation time of each of the models on the four different datasets.

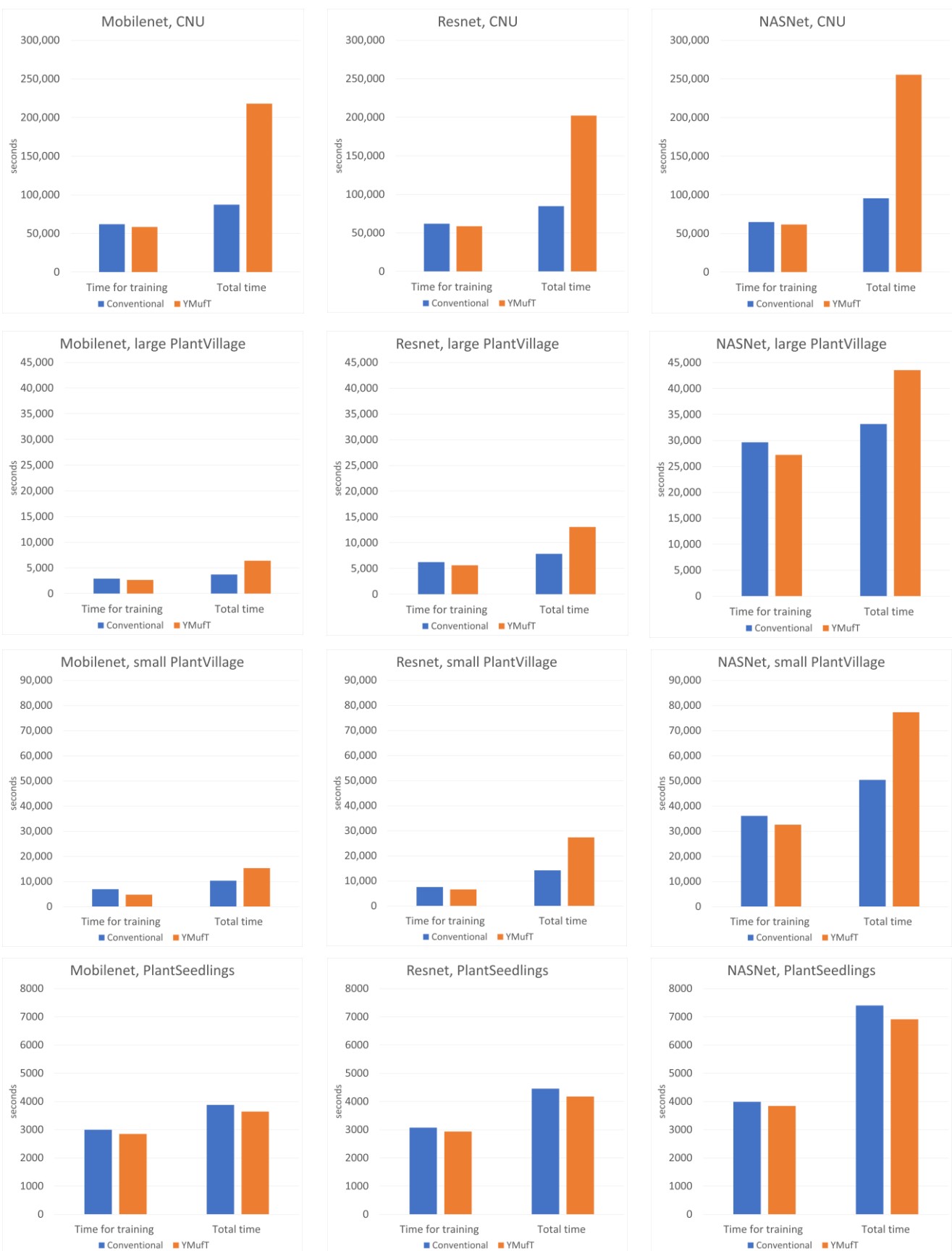

**Figure 10.** Comparison of elapsed time between the conventional method and YMufT strategy.

*5.4. Results*

5.4.1. CNU Weeds Dataset

Figure 11 shows the learning curves of the 3 models trained by the conventional and YMufT strategies. In contrast to the smooth training curve of the conventional method, the training curve of YMufT had a sawtooth appearance, which reflected the transitions between folds. When the model began to converge, sawtooth marks appeared when it learned features from samples in one fold on a few training loops but then changed to a new fold with new samples. In this case, the model first showed a drop in performance but then began generalizing on the next training loop. Notice that later folds contained many species that already appeared on previous folds, which helped the model not decrease in performance to the same extent and made it easier to converge. After a few periods, the amplitude of each sawtooth was reduced and the model started to converge throughout many folds.

The model converged at the 10th validation in the conventional strategy; this was faster than in the YMufT strategy, which required 60 validations to converge. However, as illustrated by the overall performance scores in Figure 12, use of YMufT led to a higher F1 score, while the accuracy was approximately the same as that seen with conventional training. Figure 13 shows a comparison of the precision, recall, and F1 scores by species between the YMufT and conventional training strategies. In this figure, the species are arranged in ascending order, based on the ratio of the number of samples in a species to the maximum number of samples.

On Mobilenet, although YMufT did not have a clear advantage in terms of precision, it showed improved recall on minority species over the conventional training strategy while maintaining recall on majority species. Thus, the F1 score slightly increased on minority species. In Resnet, YMufT showed an advantage in terms of precision over the conventional training strategy on minority species but remained imprecise in terms of recall. Still, the F1 score of minority species was slightly better than that of the conventional training method. In NASNet, although the overall F1 score using YMufT was marginally higher than that of the conventional strategy, recall on minority species did not improve. Only precision showed a clear advantage of YMufT on minority species, which resulted in a slight improvement in F1 score on minority species.

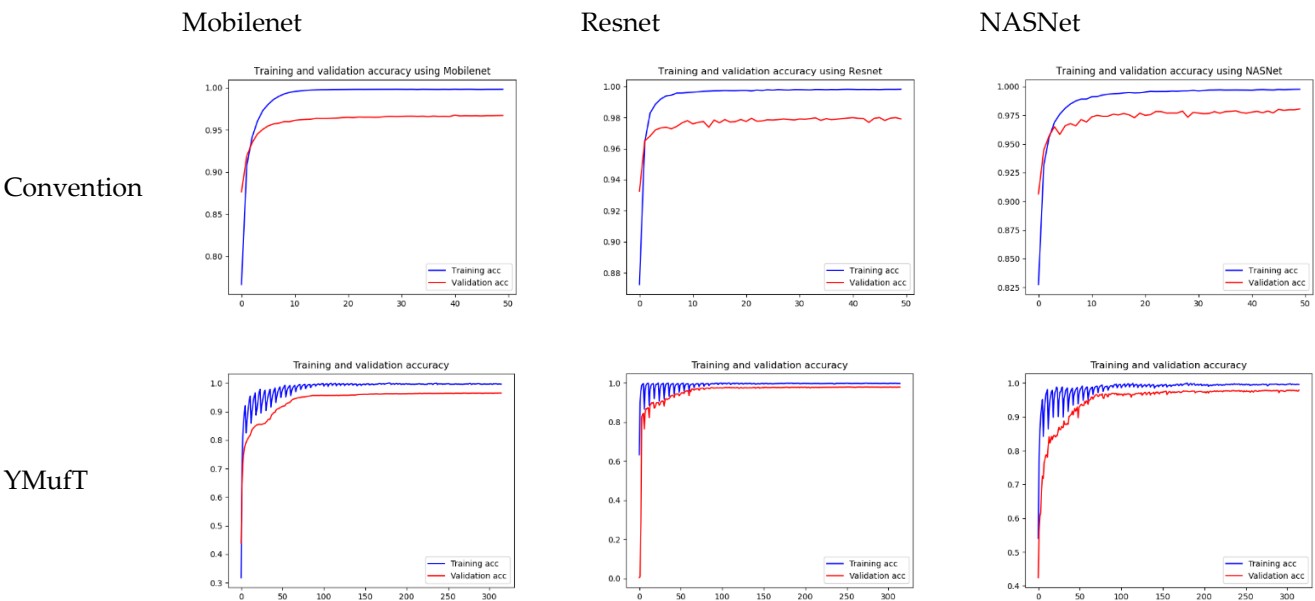

**Figure 11.** Performance of models trained on the CNU Weeds dataset using the conventional training strategy (upper row) and the YMufT strategy (lower rows).

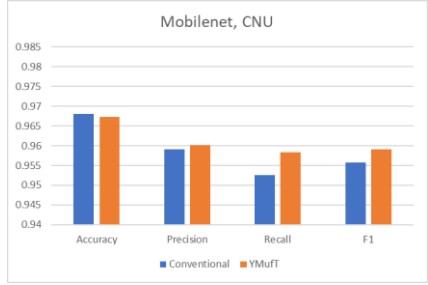 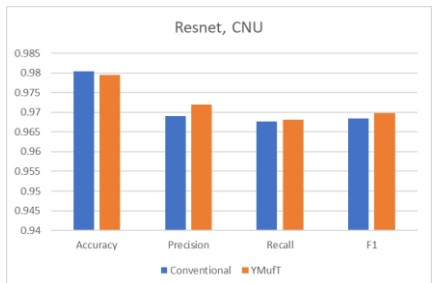 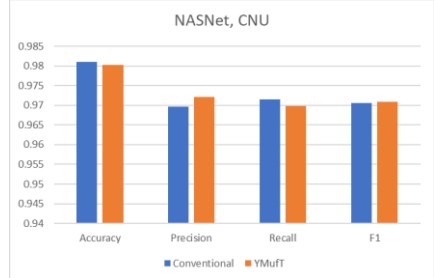

**Figure 12.** Overall performances on the CNU Weeds test set.

**Figure 13.** Precision, recall, and F1 score by species on the CNU Weeds dataset. The blue bars indicate the ratio between the number of samples of a species and the maximum number of samples in all species.

### 5.4.2. Large PlantVillage

The learning curves in Figure 14 show a shorter sawtooth than the CNU Weeds dataset because of the lack of variety in samples in the large PlantVillage dataset. In a given training fold, despite prior knowledge from the previous folds, the samples were not significantly different, which help the model converge quickly.

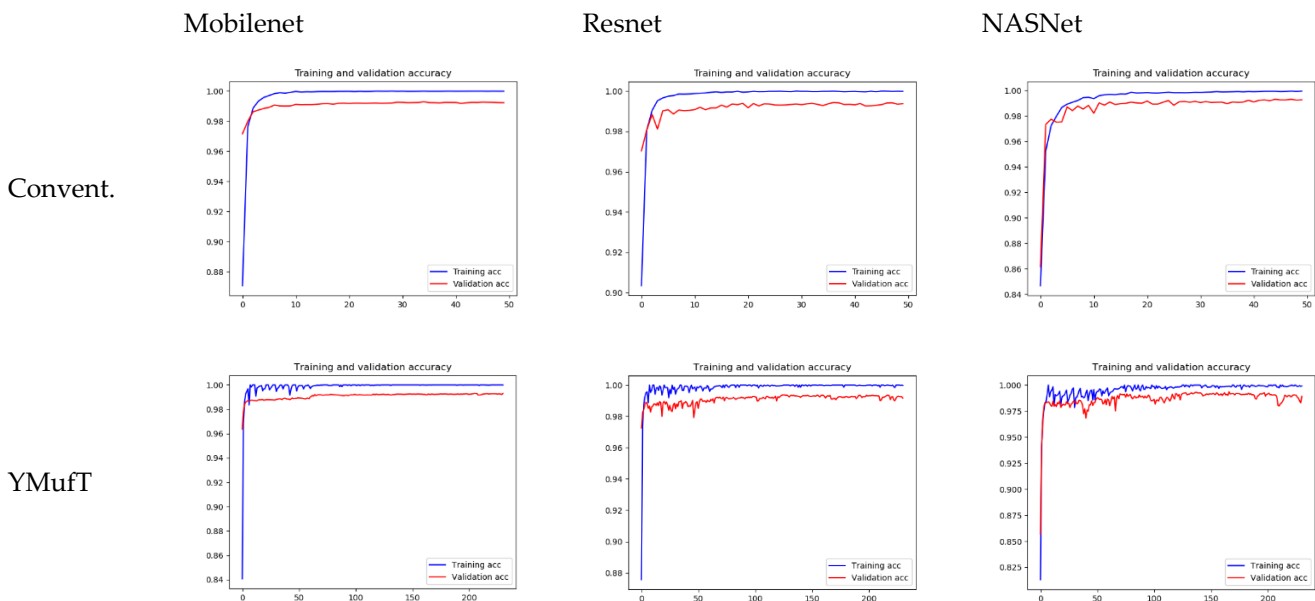

**Figure 14.** Performance when models were trained on the large PlantVillage dataset, using the conventional training (upper row) and YMufT strategies (lower rows).

As with the CNU Weeds dataset, the model converged on the first few validations faster than YMufT, which needed 60 validations to converge. Figure 15 shows the outstanding overall performance of YMufT in Mobilenet. Training Mobilenet in this dataset using the YMufT strategy resulted in an overall performance that was superior to those of Resnet and NASNet. The learning curves of the two latter models show more oscillation, illustrating the difficulty in generalizing sample characteristics in this dataset.

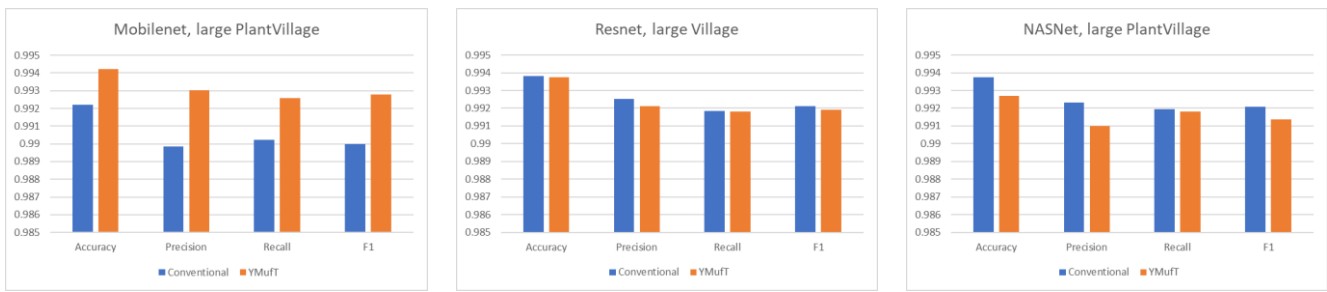

**Figure 15.** Overall performance on the large PlantVillage test set.

In Figure 16, minority classes showed a slight improvement in precision and recall using Mobilenet trained by the YMufT strategy. This improvement led to an increase in F1 score in those classes. In contrast, YMufT showed inappreciable improvement on minority classes in terms of precision and recall, making the F1 score lower than that of the conventional training method.

### 5.4.3. Small PlantVillage

Figure 17 shows the learning curves of the 3 models trained on the small PlantVillage dataset. Unlike the large dataset, the validation curves of Resnet and NASNet trained using the conventional training method showed a large degree of oscillation. Specifically, Resnet was unable to converge. In contrast, the training and validation curves of the models trained using the YMufT strategy converged quickly after 20 validation times, which was faster than those of the models trained using the conventional method. The amplitudes of the sawtooth markings were also smaller than those on the large dataset.

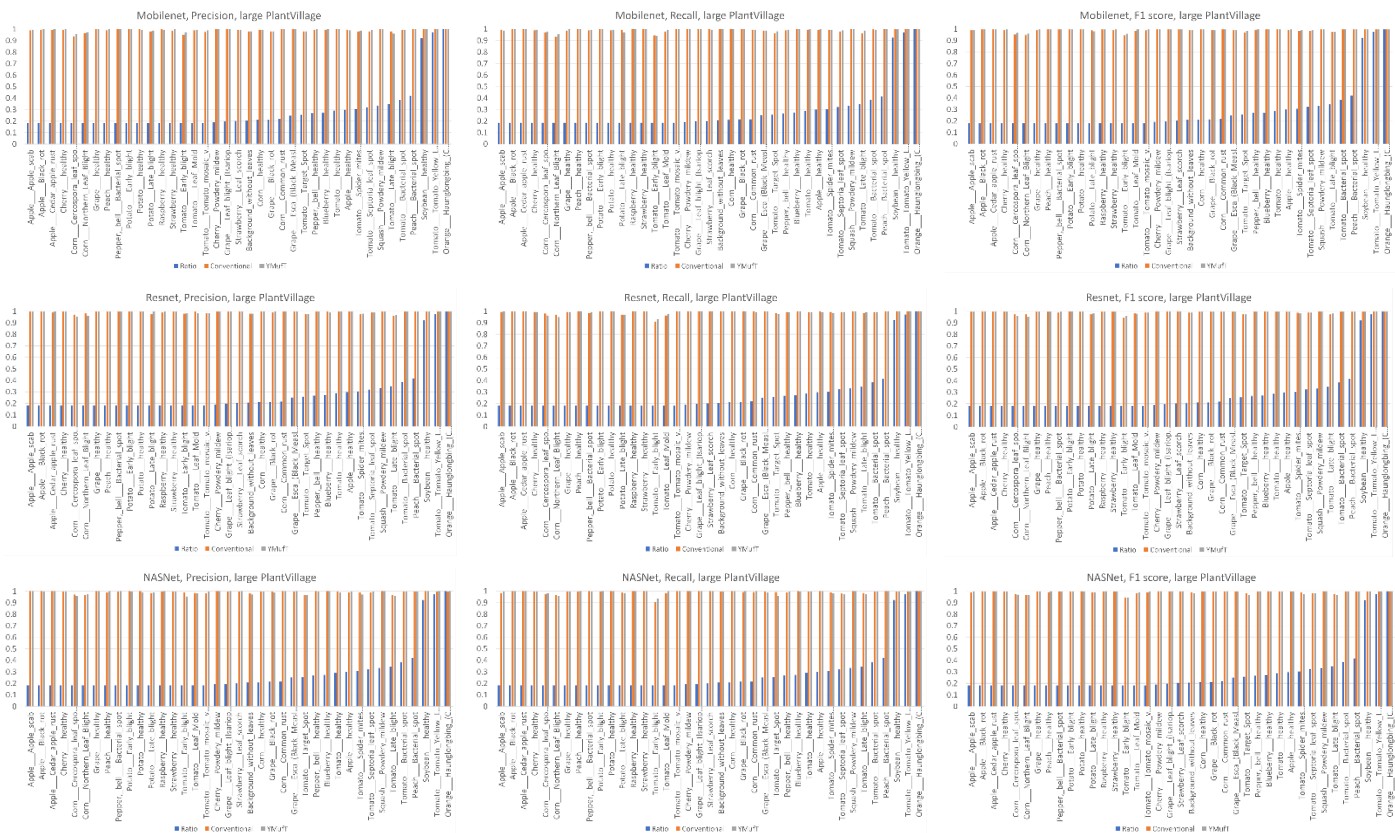

**Figure 16.** Precision, recall, and F1 score on each class in the large PlantVillage dataset. The blue bars indicate the ratio of the number of samples of a class to the maximum number of samples in all classes.

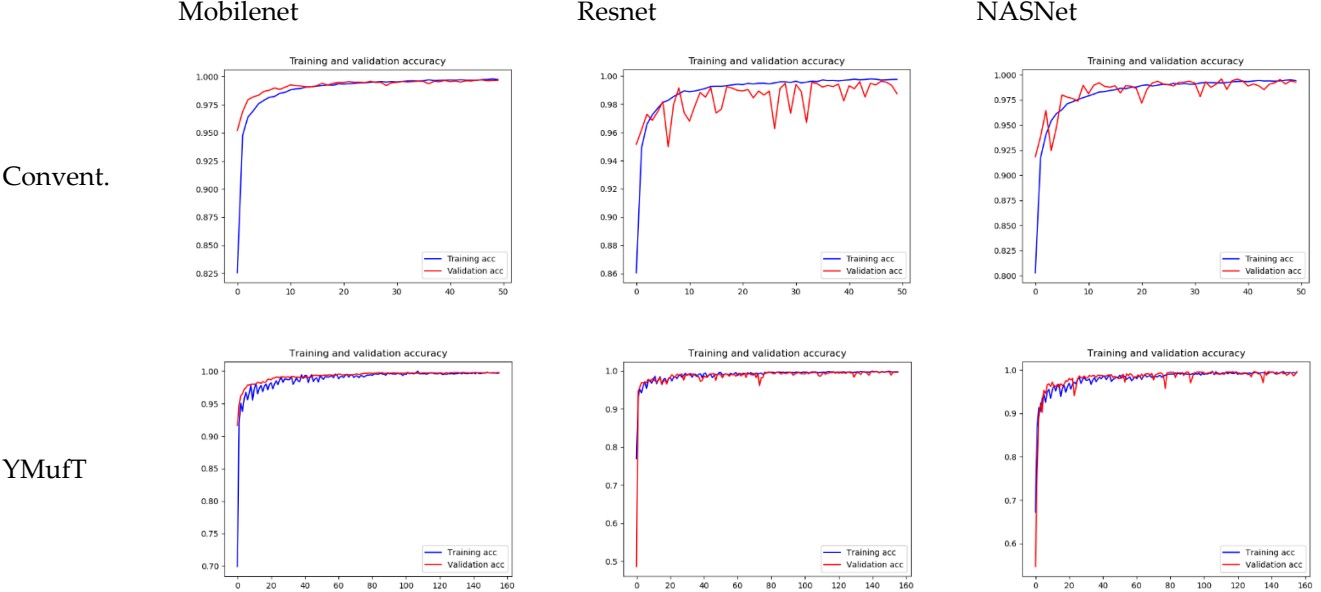

**Figure 17.** Performance of models trained on the small PlantVillage dataset, using the conventional training (upper row) and YMufT strategies (lower rows).

Figure 18 shows the average overall performance on 5-fold cross-validation. The performance of the model trained using YMufT was far superior to that of the model trained using the conventional training strategy. In general, Mobilenet and Resnet showed good performance on this dataset. As shown in Figure 19, all three models showed high

precision, recall, and F1 scores in most minority classes when the models were trained those models using the YMufT strategy.

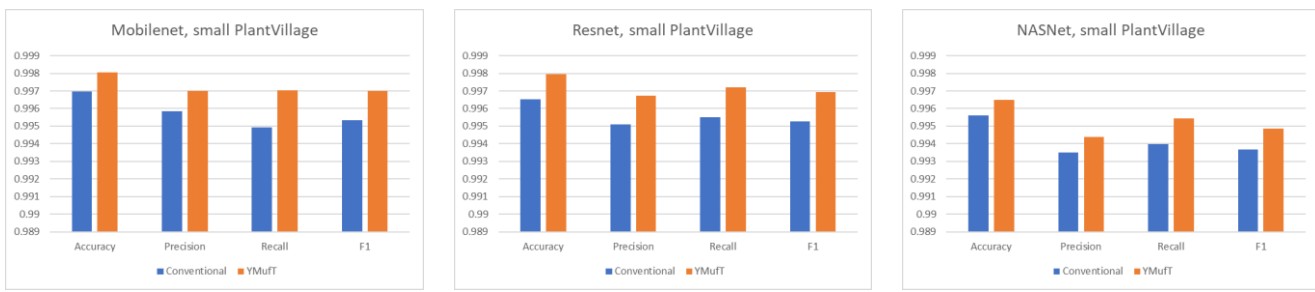

**Figure 18.** Average overall performance of 5-fold cross-validation on the small PlantVillage dataset.

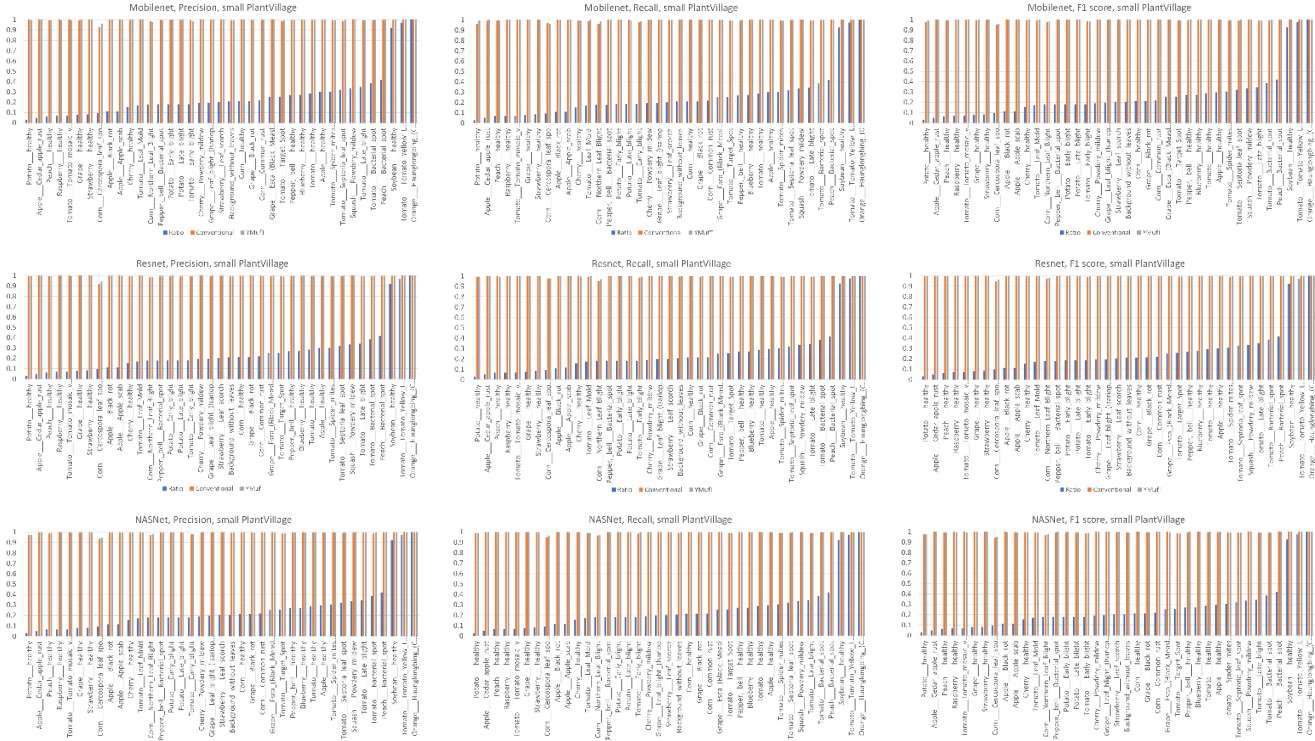

**Figure 19.** Precision, recall, and F1 score on each class in the small PlantVillage dataset. The blue bars indicate the ratio of the number of samples of a class to the maximum number of samples in all classes.

### 5.4.4. PlantSeedlings Dataset

The learning curves in Figure 20 show that both the YMufT and conventional training strategies helped the models converge quickly. In contrast, the validation curves of models trained using YMufT converged at the first few validation times, except that NASNet suffered from overfitting before the 30th validation, then quickly converged later. Like on the small PlantVillage dataset, in Figure 21, YMufT was better than the conventional training method in terms of accuracy, precision, recall, and F1 score. In Figure 22, in most cases, use of YMufT served to increase precision, recall, and F1 score in both minority and majority species.

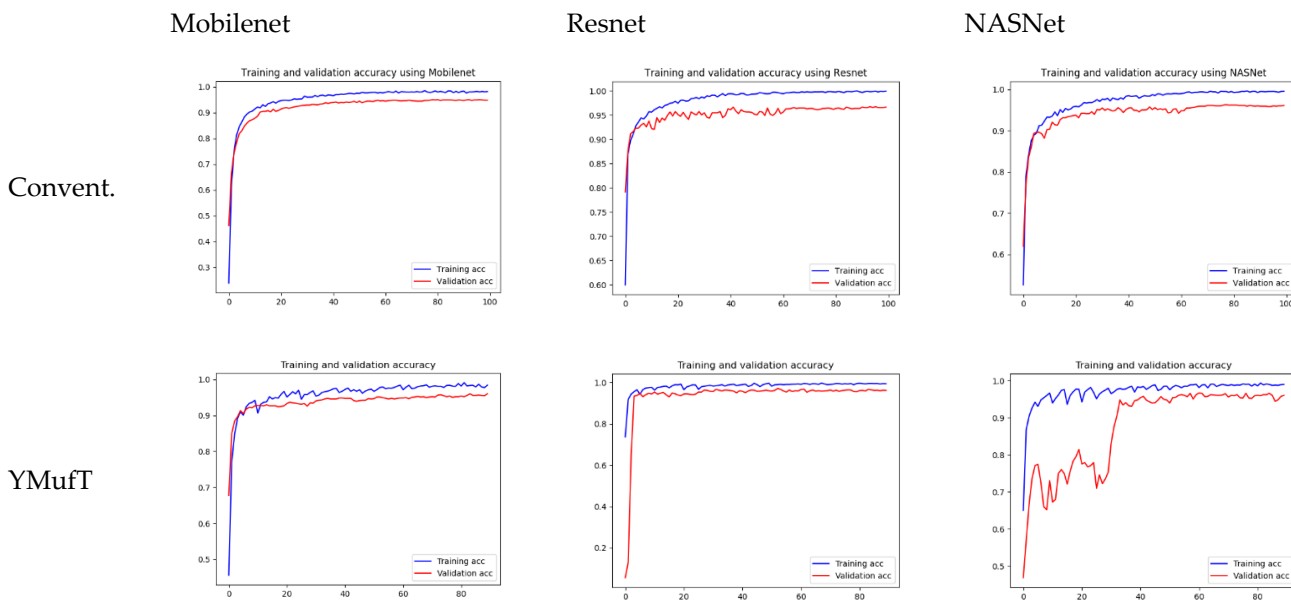

**Figure 20.** Performance of models trained on the PlantSeedlings dataset, using conventional training (upper row) and YMufT strategies (lower rows).

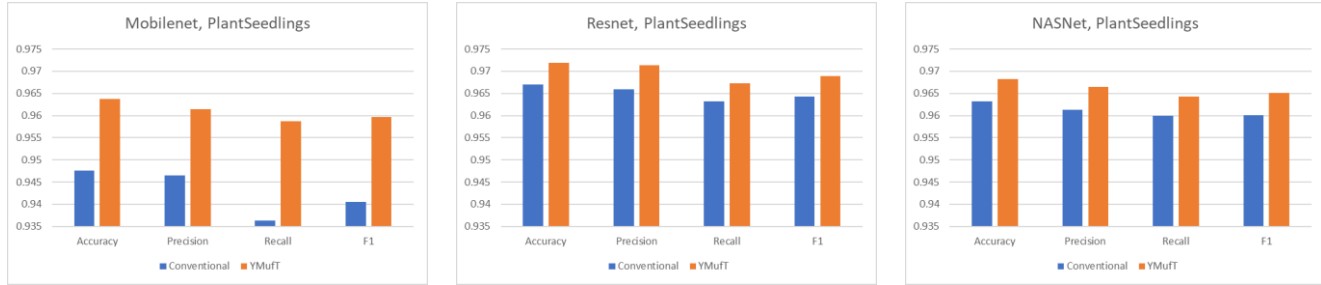

**Figure 21.** Average overall performance of 5-fold cross-validation on the PlantSeedlings dataset.

*5.5. Analysis*

Generally, on large datasets (CNU Weeds and large PlantVillage), training a model using the YMufT strategy rather than the conventional training method made the validation slower to converge. Still, the overall F1 score and the score on minor species improved, while the F1 score was maintained on major species. Of the three models, NASNet achieved the best performance on the CNU Weeds dataset, while Mobilenet was the optimal solution on the large PlantVillage dataset.

On small datasets (small PlantVillage and PlantSeedlings), the models trained using the YMufT strategy were faster to converge on validation than those trained using the conventional training method. Furthermore, using the YMufT strategy, the overall performance of the models and the performance on minor species were significantly improved in comparison to the performance of the models trained using the conventional training method. Mobilenet and Resnet were the optimal models on the small PlantVillage dataset, and Resnet achieved the highest performance on the PlantSeedlings dataset.

Table 5 compares the YMufT strategy to other methods. On the CNU Weeds dataset, the NASNet model trained using the YMufT strategy showed slightly lower performance than the other 2 DNN models. On the large PlantVillage dataset, Mobilenet trained using YMufT was the optimal solution. On the two small datasets, DNN models trained using YMufT were superior to other methods. The optimal models on the small PlantVillage dataset were Mobilenet and Resnet. Resnet was also the optimal model on the PlantSeedlings dataset.

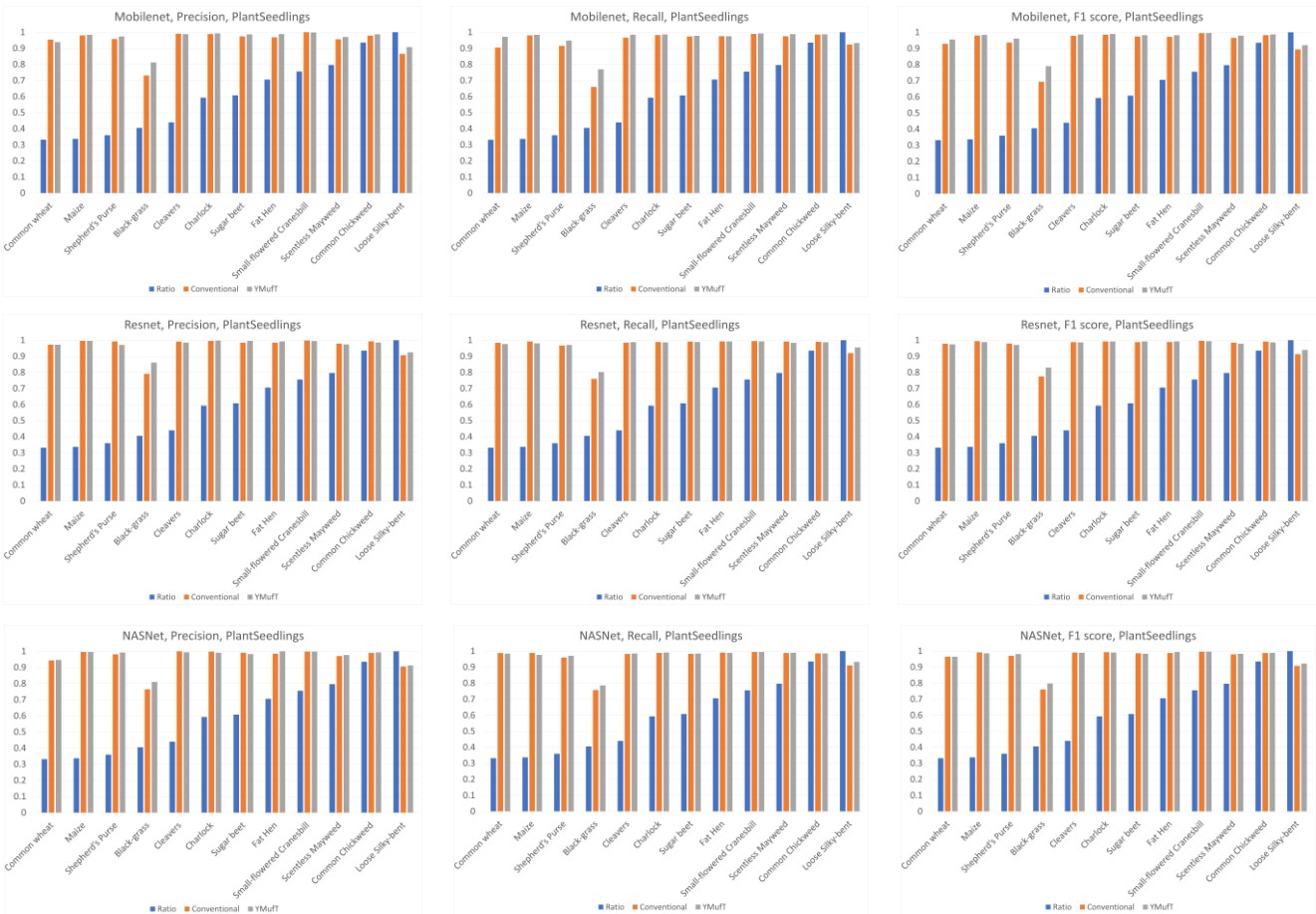

**Figure 22.** Precision, recall, and F1 score on each species in the PlantSeedlings dataset. The blue bars indicate the ratio of the number of samples of a species to the maximum number of samples in all species.

**Table 5.** Performance comparison between the YMufT strategy and other methods.

| Dataset | Method | Accuracy | Precision | Recall | F1 Score |
|---|---|---|---|---|---|
| CNU | YMufT, Mobilenet | 0.9673 | 0.9602 | 0.9583 | 0.9591 |
| | YMufT, Resnet | 0.9795 | 0.9720 | 0.9681 | 0.9698 |
| | YMufT, NASNet | 0.9802 | 0.9721 | 0.9698 | 0.9708 |
| | 2 models, product rule [23] | 0.9844 | 0.9725 | 0.9768 | 0.9746 |
| Large PlantVillage | YMufT, Mobilenet | 0.9942 | 0.9930 | 0.9926 | 0.9928 |
| | YMufT, Resnet | 0.9937 | 0.9921 | 0.9918 | 0.9919 |
| | YMufT, NASNet | 0.9927 | 0.9910 | 0.9918 | 0.9914 |
| | Convention, Mobilenet | 0.9922 | 0.9898 | 0.9902 | 0.9900 |
| | Convention, Resnet | 0.9938 | 0.9925 | 0.9918 | 0.9921 |
| | Convention, NASNet | 0.9937 | 0.9923 | 0.9920 | 0.9921 |
| Small PlantVillage | YMufT, Mobilenet | 0.9981 | 0.9970 | 0.9971 | 0.9970 |
| | YMufT, Resnet | 0.9979 | 0.9967 | 0.9973 | 0.9970 |
| | YMufT, NASNet | 0.9965 | 0.9944 | 0.9955 | 0.9949 |
| | SE-MobileNet [28] | 0.9978 | - | - | - |
| | Resnet50 [29] | 0.9959 | - | - | - |
| | DenseNet [29] | 0.9975 | - | - | - |
| | ReLU [30] | 0.9960 | - | - | - |
| | GoogleNet, transfer learning [31] | 0.9935 | 0.9935 | 0.9935 | 0.9934 |
| | ResNet34, deep [32] | 0.9967 | - | - | - |
| | Inception_v3 [32] | 0.9976 | - | - | - |

**Table 5.** *Cont.*

| Dataset | Method | Accuracy | Precision | Recall | F1 Score |
|---|---|---|---|---|---|
| PlantSeedlings | YMufT, Mobilenet | 0.9637 | 0.9614 | 0.9587 | 0.9596 |
| | YMufT, Resnet | 0.9718 | 0.9713 | 0.9672 | 0.9689 |
| | YMufT, NASNet | 0.9682 | 0.9665 | 0.9643 | 0.9652 |
| | 3 models, weighted linear Combination (per-species) [23] | 0.9704 | 0.9706 | 0.9659 | 0.9683 |
| | AgroAVNET 12 species [24] | 0.9364 | 0.9300 | 0.9400 | 0.9300 |
| | ResNet-50 [25] | 0.9621 | 0.9525 | 0.9583 | 0.9542 |
| | CFMNN [26] | 0.9110 | - | - | - |
| | ResNet 50 [27] | 0.9523 | 0.9500 | 0.9500 | 0.9500 |
| | MobileNetV2 [28] | 0.9350 | 0.9400 | 0.9400 | 0.9300 |

Figure 23 compares the number of images that were correctly classified using the conventional training method but incorrectly classified using YMufT (CtYf) and vice versa (CfYt). In all cases, use of YMufT effectively enhanced the models' ability to classify images in both minor and major species.

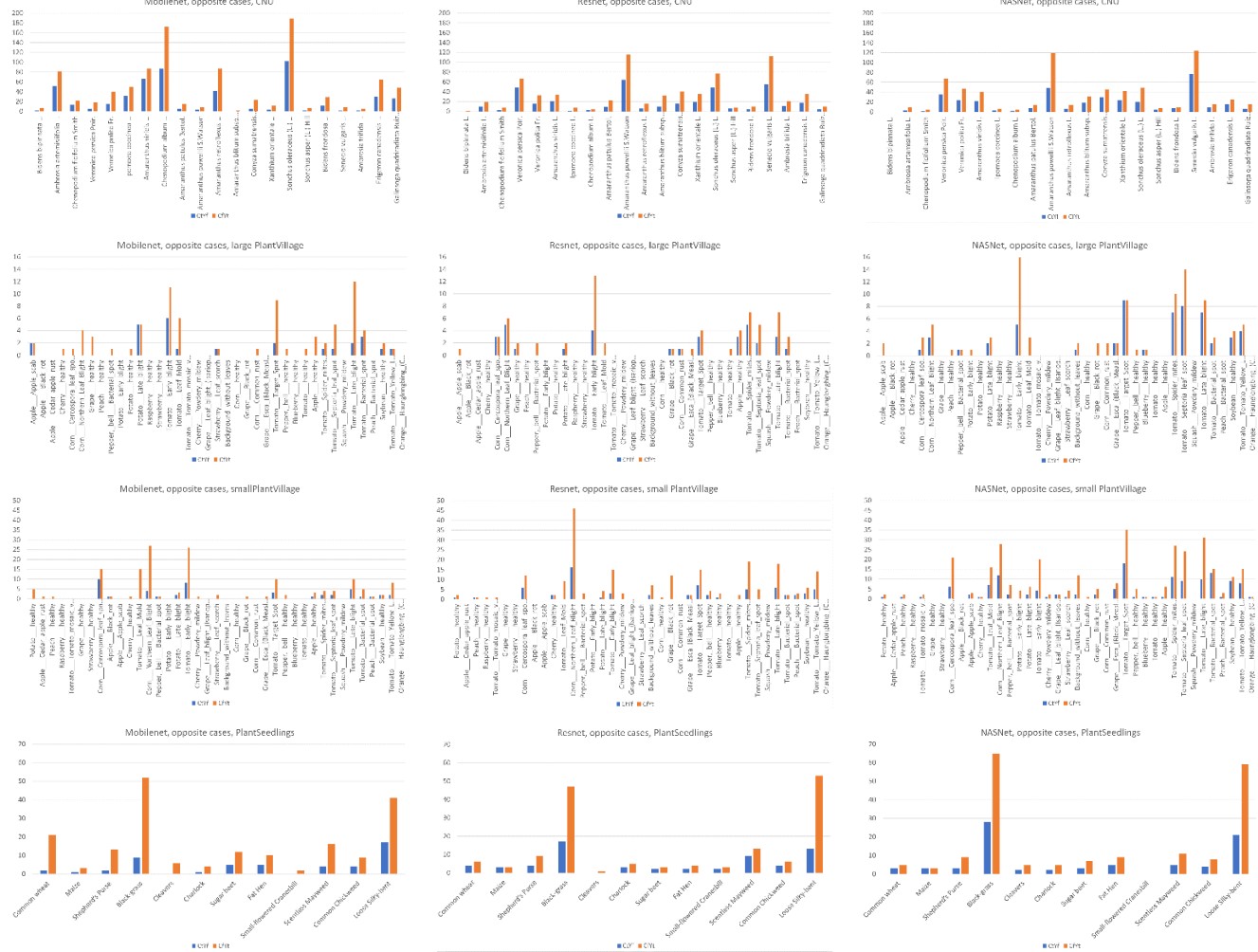

**Figure 23.** Number of images that were correctly classified using conventional training method but incorrectly classified using YMufT (CtYf), and vice versa (CfYt).

To further explore the effectiveness of YMufT, we applied Grad-CAM [41] to visualize the behavior of the models trained by the conventional method and YMufT strategy. Figure 24 shows examples of images that were incorrectly predicted by the models trained

with the conventional method but correctly classified by those trained using YMufT. On the CNU Weeds dataset, YMufT helped the models focus on the leaf surface, while conventional training was more focused on high-level features. Specifically, for the species *Veronica persica* Poir. and *Amaranthus powellii* S.Watson, conventional training focused on the lower curve position, while YMufT concentrated on leaf veins and leaf blades. A similar phenomenon also occurred on *Ambrosia artemisiifolia* L., *Ipomoea coccinea* L., and *Chenopodium album* L. species. The only image from *Chenopodium ficifolium* Smith contains overlapping leaves, so the YMufT strategy guided the model to localized areas around leaf centroids, while the conventional training method focused on a lower region. Similar things happened on the large PlantVillage dataset, such as for images of Cherry___healthy, Tomato___Late_blight, Tomato___Bacterial_spot, Apple___Apple_scab, and Potato___Early_blight. For one image of Pepper,_bell___Bacterial_spot, both methods focused on the leaf surface, but the YMufT strategy mainly concentrated on the petiole region. Generally, on large datasets, YMufT helped the model focus on the leaf surface to capture essential low-level features. In contrast, the conventional training method focused on high-level features such as leaf curves, which may not be significant in terms of species characteristics.

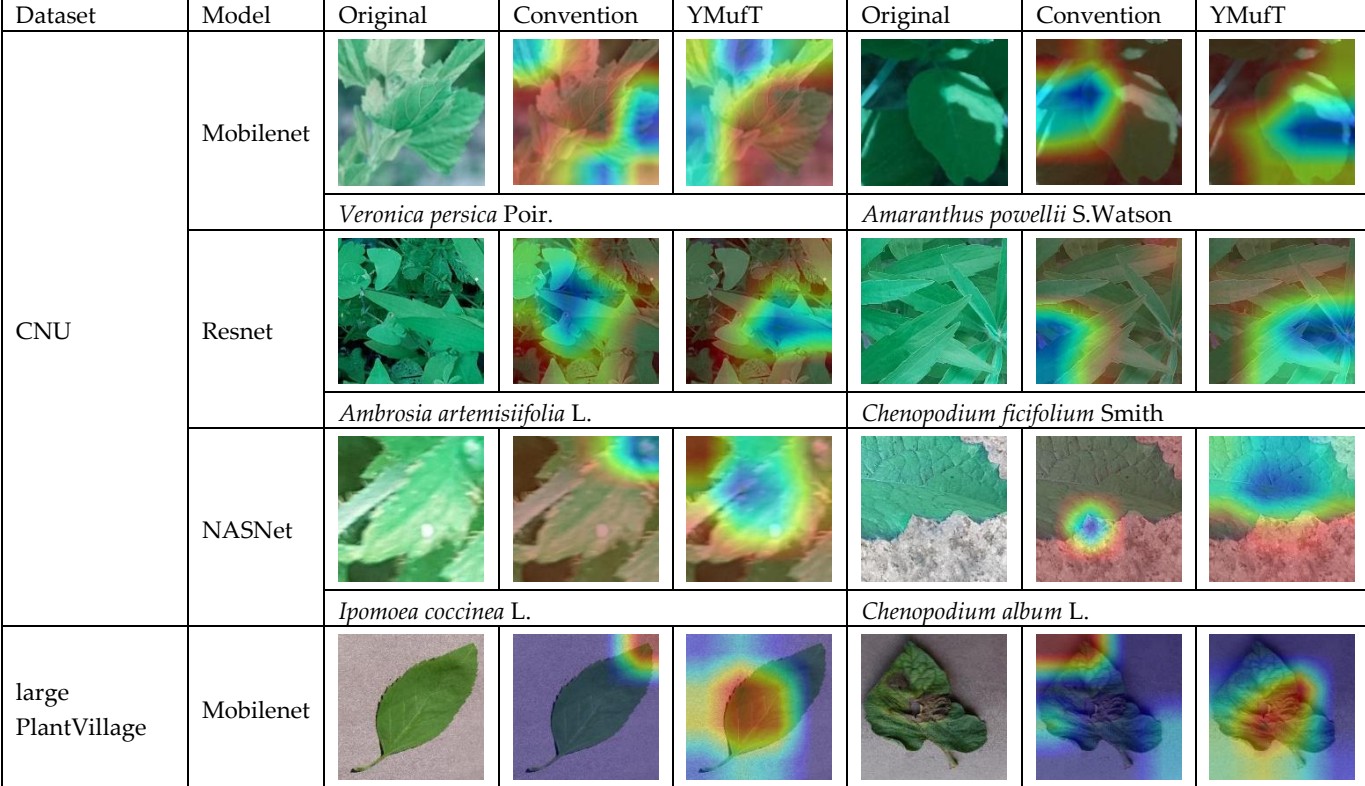

**Figure 24.** *Cont.*

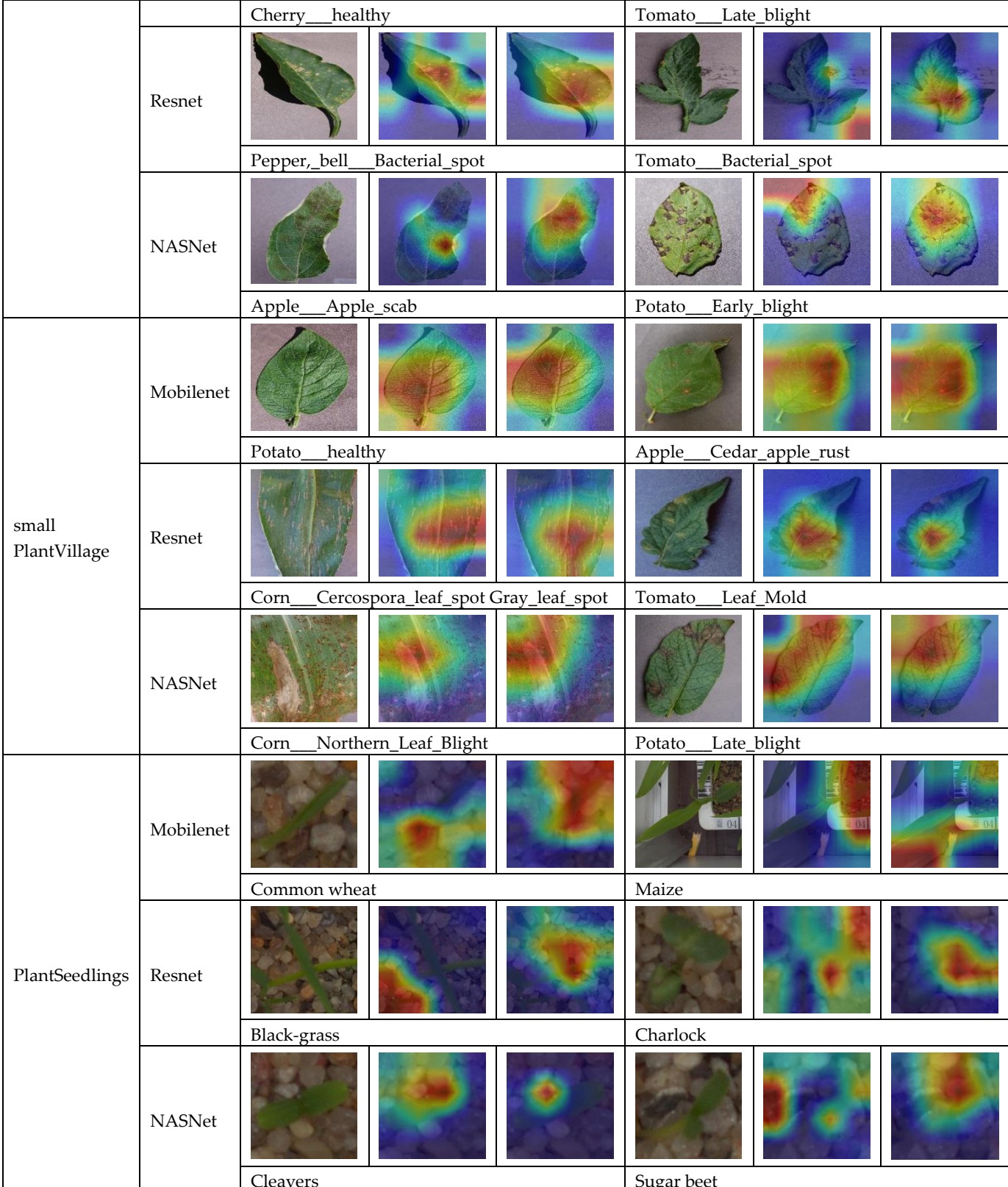

**Figure 24.** Grad-CAM visualization of models trained by the conventional method and YMufT strategy. The original images were misclassified using the conventional method but classified correctly using YMufT. Below each original image is the species name.

In contrast, both the conventional training and YMufT strategies concentrated on the surfaces in images in the small PlantVillage dataset. Still, the heatmap color shown in Potato___healthy, Apple___Cedar_apple_rust, and Tomato___Leaf_Mold indicated that YMufT made the model pay more attention to the local area in the surface and reduced its attention to patterns in the background. Other images show that the two methods displayed different feature localization behavior. The conventional training method focused on the wide region in the middle of the leaf in Corn___ Cercospora_leaf_spot Gray_leaf_spot, but the YMufT strategy looked at micro-characteristics towards the top. In Corn___Northern_Leaf_Blight, the small region near the middle was not sufficient to classify this image, so YMufT extended the region towards both sides to collect additional essential features. Consider Potato___Late_blight image: although both methods focused on the left half of the image, the conventional method mainly focused on the upper and lower curve, while YMufT considered the inner region to capture essential features. In the PlantSeedlings dataset, low-quality images, as well as the small shapes of parts of the weeds, meant that the model trained by the conventional method suffered from difficulty in localizing the target weeds. This was true for images of maize, black-grass, charlock, and sugar beet. However, the YMufT strategy guided the model so that it localized the target properly, which increased the model's performance. Examples include images of common wheat and cleavers.

## 6. Conclusions

In this work, we presented YMufT, a strategy for training DNN models on imbalanced datasets. Given an imbalanced dataset, YMufT divides the training set into multiple folds, and the model trains these folds consecutively. We proposed an MCMB procedure to divide samples from the training set into folds such that the model is trained on minority species more often than majority species, thus reducing the bias toward majority species. We developed a formula to determine the numbers of training loops and training periods. The number of times training samples are loaded in the YMufT strategy is smaller or approximately the same as that for the conventional training method. We used a sequence of decreasing consecutive natural numbers, starting with the number of the training period, as the number of training loops.

We experimented with our strategy on two large datasets (CNU and large PlantVillage) and two small datasets (small PlantVillage and PlantSeedlings). Without considering validation times that can be changed on purpose, on all types of weeds datasets, training of the model using the YMufT strategy was faster than training using the conventional training method. Despite a slight reduction in accuracy, YMufT produced an increase in the overall F1 score and the F1 score on minor species on the in-the-wild CNU weeds dataset (a large dataset). The F1 score was 0.9708 using the NASNet model. Similar results were obtained on the plain large PlantVillage weeds dataset, for which Mobilenet showed the best performance in terms of both accuracy (0.9942) and F1 score (0.9928). Use of YMufT to train DNN models on small datasets results in better model performance than use of conventional training methods. Mobilenet and Resnet were the optimal solutions for the plain small PlantVillage weeds dataset, with an accuracy of 0.9981 and F1 score of 0.9970 for Mobilenet, and an accuracy of 0.9979 and F1 score of 0.9970 for Resnet. Resnet was also the best-performing model on the in-the-wild PlantSeedlings dataset, with an accuracy of 0.9718 and an F1 score of 0.9689.

We used Grad-CAM to visualize and analyze the models' behavior on large datasets. YMufT guided the model to focus on learning essential features on the leaf surfaces, while conventional training method led the model to pay attention to high-level features such as leaf curves or leaf centroids, which might be insufficient to describe species characteristics. On the small PlantVillage weeds dataset, both approaches concentrated on the leaf surface. Still, YMufT made the model pay more attention to the local area on the surface and reduced capture of patterns in the background. On the PlantSeedlings dataset, YMufT guided the model to properly localize the weeds targets.

**Author Contributions:** Conceptualization: P.T.B.; methodology: P.T.B., coding: V.H.T.; validation: V.H.T.; formal analysis: P.T.B. and K.J.Y.; investigation: V.H.T.; writing: V.H.T., writing review: P.T.B. and K.J.Y.; supervision: P.T.B. and K.J.Y.; project administration: K.J.Y., funding acquisition: K.J.Y. and Y.G.H., Data curation (CNU dataset): Y.G.H. All authors have read and agreed to the published version of the manuscript.

**Funding:** This research was funded by the "Cooperative Research Program for Agriculture Science and Technology Development (Project No. PJ01385501)," Rural Development Administration, Republic of Korea.

**Institutional Review Board Statement:** Not applicable.

**Informed Consent Statement:** Not applicable.

**Data Availability Statement:** PlantVillage (large and small) [37], available at https://data.mendeley.com/datasets/tywbtsjrjv/1 (accessed on 6 April 2021). PlantSeedlings [36], available at https://vision.eng.au.dk/plant-seedlings-dataset/ (accessed on 6 April 2021).

**Acknowledgments:** This work was carried out with the support of the "Cooperative Research Program for Agriculture Science and Technology Development (Project No. PJ01385501)," Rural Development Administration, Republic of Korea.

**Conflicts of Interest:** The authors declare no conflict of interest.

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
