# Peer review of "Yielding Multi-Fold Training Strategy for Image Classification of Imbalanced Weeds"

_applsci, doi:10.3390/app11083331_

Round 1
Reviewer 1 Report
Content
----------
The goal of this paper is to proposes a Yielding Multi-fold Training (YMufT) strategy to train a DNN model on an imbalanced dataset. This strategy reduces the bias problem on training by a Min-Class-Max-Bound procedure (MCMB), dividing samples on the training set into multiple folds. Furthermore, the proposed strategy has been trained on two small (PlantSeedlings, small PlantVillage) and two large (Chonnam National University (CNU), large PlantVillage) weeds datasets. The author also discussed that YMufT guides the model to focus and learn features on the leaf surface, while conventional training methods mainly concentrate on high-level features.
Major comments
--------------
1. Data set from real world
Two small weeds datasets (Plant Seedlings and dataset and non-augment version of Plant Village dataset) and two large (Chonnam National University (CNU) weeds dataset and augment version of Plant Village dataset).
What is the differnce for these four data sets? What is the purpose for each data set?
2. Missing reference
Ramazan Gokberk Cinbis, Jakob Verbeek, Cordelia Schmid. Multi-fold MIL Training forWeakly Supervised Object Localization. CVPR 2014 - IEEE Conference on Computer Vision& Pattern Recognition, Jun 2014, Columbus, United States. IEEE, 2014.
Evaluation
--------------
Given the above, I'm in a position to major revision.
Author Response
- Data set from the real world
Two small weeds datasets (Plant Seedlings and dataset and non-augment version of Plant Village dataset) and two large (Chonnam National University (CNU) weeds datasets and augment version of Plant Village dataset).
What is the difference between these four data sets? What is the purpose of each data set?
→ I added the explanation from Line 164 – 173. In these four datasets, CNU and PlantSeedlings are in-the-wild datasets, small and large PlantVillage datasets are the plain datasets. But CNU and large PlantVillage are large datasets, and the others are small datasets. The various datasets help us analyze the benefit of YMufT on different types of datasets.
- Missing reference
Ramazan Gokberk Cinbis, Jakob Verbeek, Cordelia Schmid. Multi-fold MIL Training forWeakly Supervised Object Localization. CVPR 2014 - IEEE Conference on Computer Vision& Pattern Recognition, Jun 2014, Columbus, United States. IEEE, 2014.
→ I added on Line 35.
Reviewer 2 Report
The paper presents a method to train a DeepNeural Network model on an imbalanced dataset. The performance of the method is tested on different sizes weeds data sets.
Comment 1: Some Figures are not readable (texts too small) Figures 13, 16, 19 and specially 23 have to be changed.
Comment 2: In several equations use the cross product instead of the scalar product. One example is Eq. 20.
Comment 3: Conclusion section should be extended highlighting the result and analysis subsection.
Author Response
Comment 1: Some Figures are not readable (texts too small) Figures 13, 16, 19 and specially 23 have to be changed.
→ I changed. Also, in the submission package, I attach high-resolution images for further editions.
Comment 2: In several equations use the cross product instead of the scalar product. One example is Eq. 20.
→ I changed all cross-product notations into dot product notations.
Comment 3: Conclusion section should be extended highlighting the result and analysis subsection.
→ I extend results and write further analysis, such as the best performance in each dataset (Line 624 - 634), models’ behavior using Grad-CAM (Line 635 - 642).
Reviewer 3 Report
The aim of the paper entitled Yielding Multi-Fold Training Strategy for Imbalanced Weeds Image Classification is topresent a new way to train a DNN model. Overall paper is interesting but there are a few parts which should improved.
Please add results in the abstract.
Please write the full name of DNN in the introduction. The introduction is very difficult to understand. The authors cite a lot of literature, but the fact that the citations are placed in parentheses disturbs the reception of the text. Please indicate some authors and the year of publication. Please add any flowchart of similar research in the first chapter.
Please improve the resolution of charts.
Please extend the conclusion section.
Author Response
Please add results in the abstract.
→ I extend results by informing the best performance and corresponding value in each dataset (Line 21-24), models’ behavior using Grad-CAM (Line 25-28).
Please write the full name of DNN in the introduction.
→ In Line 32, I rewrite ‘DNN’ into ‘Deep Neural Network (DNN)’.
The introduction is very difficult to understand. The authors cite a lot of literature, but the fact that the citations are placed in parentheses disturbs the reception of the text. Please indicate some authors and the year of publication. Please add any flowchart of similar research in the first chapter.
→ I shorten continuous citation indexes into short forms, such as [5][6][7][8][9] in Line 37 into [5-9]. I write authors’ names in a single index, such as rewrite ‘[10] mentioned ...’ in Line 38 into ‘Cao et al. mentioned ...’.
Please improve the resolution of charts.
→ I changed. Also, in the submission package, I attach high-resolution images for further editions.
Please extend the conclusion section.
→ I extend results and write further analysis, such as the best performance in each dataset (Line 624 - 634), models’ behavior using Grad-CAM (Line 635 - 642).
Round 2
Reviewer 1 Report
The author revised as previous comments.